# Compiler-R1: Towards Agentic Compiler Auto-tuning with Reinforcement Learning

**Haolin Pan**[1,2,3]*, **Hongyu Lin**[2,3]*, **Haoran Luo**[4,†], **Yang Liu**[2,3]
**Kaichun Yao**[2], **Libo Zhang**[2], **Mingjie Xing**[2,3,†], **Yanjun Wu**[2,3]
[1]Hangzhou Institute for Advanced Study, UCAS, China
[2]Institute of Software Chinese Academy of Sciences, China
[3]University of Chinese Academy of Sciences, China
[4]Nanyang Technological University, Singapore
{hongyu2021,mingjie}@iscas.ac.cn, haoran.luo@ieee.org
panhaolin21@mails.ucas.ac.cn

## Abstract

Compiler auto-tuning optimizes pass sequences to improve performance metrics such as Intermediate Representation (IR) instruction count. Although recent advances leveraging Large Language Models (LLMs) have shown promise in automating compiler tuning, two significant challenges still remain: the absence of high-quality reasoning datasets for agents training, and limited effective interactions with the compilation environment. In this work, we introduce Compiler-R1, the first reinforcement learning (RL)-driven framework specifically augmenting LLM capabilities for compiler auto-tuning. Compiler-R1 features a curated, high-quality reasoning dataset and a novel two-stage end-to-end RL training pipeline, enabling efficient environment exploration and learning through an outcome-based reward. Extensive experiments across seven datasets demonstrate Compiler-R1 achieving an average 8.46% IR instruction count reduction compared to `opt -Oz`, showcasing the strong potential of RL-trained LLMs for compiler optimization. Our code and datasets are publicly available at `https://github.com/Panhaolin2001/Compiler-R1`.

## 1 Introduction

Compiler auto-tuning [2, 5, 6] focus on automatically select and order compilation passes, modular optimization or analysis steps in modern compilers like LLVM [20], to improve program performance. A typical tuning objective is reducing the Intermediate Representation (IR) instruction count while preserving program correctness, as depicted in Figure 1. However, this task presents inherent challenging due to the combinatorial explosion of possible pass sequences and the intricate interactions between individual passes.

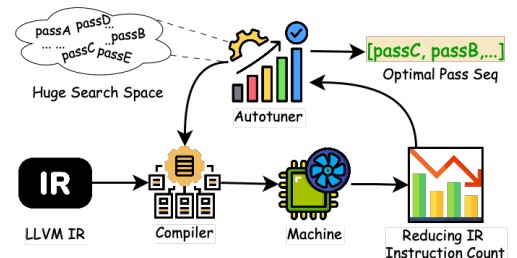

Figure 1: Overview of compiler auto-tuning task.

Recent years have witnessed significant advancements in compiler auto-tuning methods, evolving through three major generations: **Heuristic-based Methods**: Early work in iterative compilation [4, 7, 13, 1] establish fundamental search strategies

---

*These authors contributed equally to this work.
†Corresponding author(s).

39th Conference on Neural Information Processing Systems (NeurIPS 2025).

such as genetic algorithm and simulated annealing. **Machine Learning (ML)-enhanced Methods**: Subsequent approaches [6, 31, 15, 26] leveraged ML models to improve tuning efficiency on unseen benchmarks. However, these traditional approaches, including both heuristic and ML methods, often suffer from inherent inefficiency and limited generalization. **Large Language Models (LLMs)-based Methods**: With the advent of LLMs, recent efforts [17, 9, 8] have explored their application to compiler auto-tuning tasks, mitigating some of the aforementioned limitations.

Although recent LLMs [11, 18] exhibit remarkable reasoning and problem-solving capabilities, directly applying them to compiler auto-tuning faces two critical challenges. First, there is an **absence of high-quality training datasets tailored for auto-tuning**. Effective interactive learning for LLM agents requires datasets that include Chain-of-Thought (CoT) reasoning, external tool integration, and environment feedback, which are currently lacking in compiler tuning domains. Second, existing methods based on Supervised Fine-Tuning (SFT) exhibit **limited interaction with the compilation environment**, restricting the model's ability to autonomously explore and adaptively learn from feedback, thus resulting in poor generalization to unseen programs.

To address these challenges, we propose **Compiler-R1**, the first reinforcement learning (RL)-based framework specifically designed to enhance LLM capabilities in compiler auto-tuning. Compiler-R1 introduces two main innovations: **(1) A high-quality auto-tuning reasoning dataset:** We curate a comprehensive dataset comprising 19,603 carefully constructed samples, encompassing diverse and representative tuning scenarios, enabling effective interactive learning for compiler optimization tasks. **(2) A two-stage end-to-end RL training framework:** We develop a novel RL framework that equips LLM agents with tool interfaces for autonomous environment exploration and leverages an outcome-based reward function to guide learning. This approach enables the agent to capture both individual pass characteristics and their compositional interactions, significantly enhancing generalization capability and tuning efficiency.

We validate Compiler-R1 using various open-source LLM backbones on seven representative benchmarks. Experimental results indicate that Compiler-R1 consistently achieves an average IR instruction count reduction of 8.46% compared to the baseline method (`opt -Oz`), demonstrating the considerable promise of RL-trained LLMs for advancing compiler auto-tuning research.

Our principal contributions are summarized as follows:

- We construct a comprehensive, high-quality auto-tuning reasoning dataset by rigorously selecting and integrating multiple authoritative compiler datasets, encompassing diverse and representative tuning scenarios.

- We propose **Compiler-R1**, the first RL-based two-stage training framework specifically designed to augment LLMs with tool-invocation and advanced reasoning capabilities tailored to compiler auto-tuning tasks.

- We demonstrate through comprehensive evaluations that **Compiler-R1** consistently outperforms other baseline methods, highlighting the substantial promise of RL-enhanced LLM approaches for advancing compiler optimization.

## 2 Related Work

### 2.1 Rule-based Reinforcement Learning with Large Language Models

Reinforcement Learning (RL) enables agents to perform sequential decision-making by interacting with environments to maximize cumulative rewards. Recent advancements in rule-based RL frameworks, such as DeepSeek-R1 [11], have notably enhanced the reasoning and decision-making capabilities of Large Language Models (LLMs). This approach has effectively been extended to various domains, including question answering [25], knowledge retrieval [19, 24], logical reasoning [29], mathematical problem-solving [28], code generation [23], and operating system [22]. Nevertheless, the application of rule-based RL to compiler auto-tuning tasks remains largely unexplored.

### 2.2 Compiler Auto-tuning

Compiler auto-tuning aims to automatically identify optimal pass sequences to enhance program performance. Traditional compilers like GCC and LLVM utilize fixed optimization pipelines designed

by domain experts, which often do not adapt well to specific programs or hardware. Iterative compilation methods [7, 13] employing heuristics such as genetic algorithms incur substantial overhead. Supervised learning approaches, like coreset-NVP [21], rely heavily on labeled optimal sequences, which are challenging to obtain. RL-based methods, such as AutoPhase [15], dynamically adjust optimization strategies but have limited generalization. Recently, LLM-based approaches [8] have shown potential by directly generating pass sequences from source code; however, they typically lack sufficient environmental interaction, constraining their reasoning and adaptability.

## 3   Methodology: Compiler-R1

We present Compiler-R1, a novel framework that introduces large language model (LLM)-driven compiler auto-tuning, incorporating both supervised learning and reinforcement learning (RL) techniques. As illustrated in Figure 2, Compiler-R1 represents the first application of reinforcement learning to LLM-based compiler optimization, enabling models to move beyond imitation and autonomously learn optimization strategies through structured interaction with a simulated compilation environment.

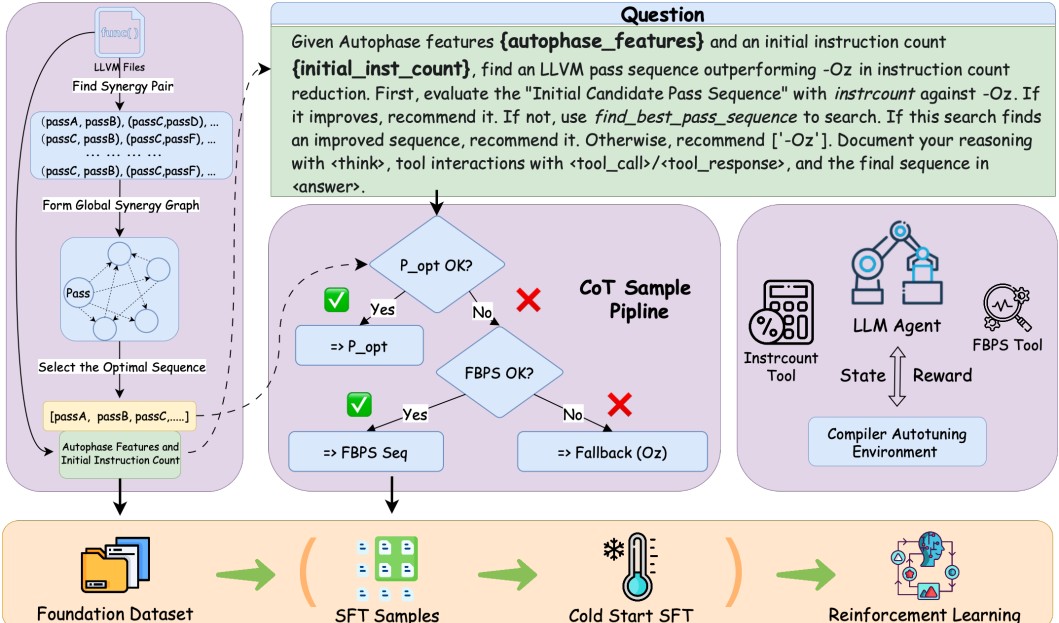

Figure 2: Compiler-R1: A two-stage LLM training framework for compiler auto-tuning. **Stage 1 (SFT):** A foundation dataset derived from synergistic pass analysis is used to create SFT samples. These samples guide the LLM through a CoT process, teaching interaction with `instrcount` and `find_best_pass_sequence` (FBPS) tools. **Stage 2 (RL):** The SFT-initialized LLM agent interacts with the compiler autotuning environment, receiving state feedback and rewards to learn an optimal policy for pass sequence generation.

### 3.1   Foundation Dataset Construction

To enable effective compiler auto-tuning, we construct a high-quality dataset tailored for LLM training, derived from empirical compiler tuning workflows (e.g., CFSAT [26]) and enhanced with structural insights. Our process involves the following pipelines:

**Raw LLVM IR Dataset Preparation.** We aggregate multiple public datasets from CompilerGym [10], applying coreset-inspired sampling [21] to ensure representational diversity. Programs with more than 10,000 IR instructions are excluded for computational feasibility. The training set comprises six uncurated datasets, while four curated benchmarks (cbench-v1, chstone-v0, mibench-v1, and npb-v0 [12, 14, 3, 16]) are served for out-of-distribution evaluation.

**Synergy Pass Pair Identification.** For each training program $P$, we derived its high-quality pass sequences $S_P$ by collecting synergistic optimization pass pairs $(A, B) \in O \times O$ satisfying the following criteria:

$$S_P = \{(A, B) \mid (\text{Count}(\text{Apply}(P, B)) < \text{Count}(P)) \wedge$$
$$(\text{Count}(\text{Apply}(P, A, B)) < \text{Count}(\text{Apply}(P, B)))\} \tag{1}$$

where $\text{Apply}(P, B)$ denotes applying pass $B$ to program $P$, and $\text{Count}(\cdot)$ is the IR instruction count. A pair $(A, B)$ is considered if $B$ alone reduces the instruction count and $A$ further reduces it when applied before $B$. These synergistic pairs indicate potential beneficial interactions between passes on a per-program basis.

**Global Synergy Graph Construction.** We aggregate these per-program synergistic pass pairs to construct a Global Synergy Graph, where nodes represent passes and edges denote pairs $(A, B)$ found to be synergistic for at least one program. This global perspective helps identify recurring beneficial pass interactions across the dataset.

**Graph-Guided Optimal Sequence Selection.** Drawing from the Global Synergy Graph, we generate 100 candidate pass sequences ($S_{cand}(P)$) per program using random walks that prioritize synergistic combinations. Each candidate $s \in S_{cand}(P)$ is evaluated by its relative improvement over the $\text{Oz}$ baseline, quantified as:

$$\text{OverOz}(s, P) = \frac{\text{Count}(\text{Apply}(P, \text{Oz})) - \text{Count}(\text{Apply}(P, s))}{\text{Count}(\text{Apply}(P, \text{Oz}))} \tag{2}$$

The sequence maximizing $\text{OverOz}$ is selected as the optimal pass sequence $\text{PassSeq}_{\text{opt}}(P)$. This empirical, graph-informed method, validated against a strong baseline, provides high-quality optimization examples for training. The derived $\text{PassSeq}_{\text{opt}}(P)$ for each program then serves as the target solution used to construct the **Simulated LLM Thought and Action Trajectory** for the SFT stage of our training pipeline.

**Feature Extraction and Representation.** To address LLM context limits, we compress programs representations using AutoPhase's 56 statistical features [15], capturing structural properties such as type distributions and control-flow properties. The extracted **AutoPhase features**, along with the program's **initial instruction count**, form the core components of the "question" part of our SFT samples, representing the program to the LLM.

## 3.2 Training Pipeline

We propose a two-stage training pipeline designed to progressively enhance LLMs' capabilities in auto-tuning task. The first stage employs supervised fine-tuning (SFT) to establish fundamental reasoning and tool invocation skills. The second stage utilizes applies RL to refine responses based on performance feedback.

### 3.2.1 Stage 1: Supervised Fine-Tuning for Cold Start

The SFT stage addresses the cold-start challenge by teaching the LLM how to reason over program features and interact with external tools to construct optimized pass sequences. raining data is structured to emulate interactive problem-solving, following a "thought–action–feedback" loop that mirrors realistic tool use.

**Core External Tools for SFT.** Two key tools are integrated into the SFT process:

- `instrcount`: Evaluates a candidate pass sequence by computing its improvement over the standard $\text{Oz}$ optimization, based on IR instruction count reduction.
- `find_best_pass_sequence`: Invoked when the initial candidate is ineffective ($\text{OverOz} \leq 0$), this tool searches for a better sequence using the guided search algorithm.

**SFT Sample Construction.** SFT training samples are designed to simulate interactive problem-solving sessions, each comprising a structured input-output pair. The input represents the problem formulation, including the target program's AutoPhase features, a clear task instruction (e.g., "find an IR-reducing pass sequence"), and the initial IR instruction count. The output models the LLM's step-by-step reasoning and decision-making process, structured as a Chain-of-Thought (CoT) trajectory.

**Simulated LLM Thought and Action Trajectory:** This part serves as the model's "answer," demonstrating a step-by-step reasoning process. It begins with a `<think>` block where the LLM analyzes the optimal pass sequence $\text{PassSeq}_{\text{opt}}(P)$ and plans to verify its effectiveness using the `instrcount` tool. A corresponding `<tool_call>` is issued, followed by a simulated `<tool_response>` containing execution status and the $\text{OverOz}$ metric. The LLM then processes this feedback in another `<think>` block to decide whether to accept the candidate or explore alternatives. This entire reasoning flow, structured with tags like `<think>`, `<tool_call>`, `<tool_response>`, and `<answer>`, forms a complete and learnable SFT sample.

---

**Training Prompt for Compiler-R1**

Act as a compiler optimization expert finding an optimal pass sequence for LLVM IR, aiming to reduce the total instruction count. The LLVM IR code is represented by statistical features. The initial statistical features are: **{autophase_features}**. Initial instruction count: **{initial_inst_count}**. Note: When calling the 'instrcount' and 'find_best_pass_sequence' tools, use the exact filepath: {filepath}

Your task is to: Provide and evaluate the Initial Candidate Pass Sequence using the `instrcount` tool to determine its instruction count improvement compared to the default `-Oz` optimization. If the initial sequence provides a positive improvement (`improvement_over_oz > 0`), recommend it as the final answer. If the initial sequence does not provide a positive improvement (`improvement_over_oz <= 0`), use the `find_best_pass_sequence` tool to search for a better sequence. If the search finds a sequence with positive improvement (`improvement_percentage > 0`), recommend that sequence. If the search tool fails to find a sequence with positive improvement, recommend the default `['-Oz']` sequence as the safest option.

Present your reasoning step-by-step using <think></think> tags and tool interactions using <tool_call></tool_call> and <tool_response></tool_response> structure, concluding with the final recommended sequence in an <answer></answer> tag.

---

### 3.2.2 Stage 2: Reinforcement Learning for Policy Optimization

After the SFT-based initialization, we further optimize the LLM's policy through reinforcement learning (RL). This stage aims to move beyond the imitative behavior learned via SFT, enabling the LLM—now acting as an agent—to autonomously discover more efficient optimization strategies through trial-and-error interactions with a simulated compilation environment. The agent's actions include generating internal reasoning traces, determining when and how to invoke external tools, and producing the final optimization sequence.

**Agent-Environment Interaction.** The LLM interacts with the environment by emitting token sequences that correspond to high-level actions, such as `<think>` statements, `<tool_call>` invocations (e.g., `instrcount`, `find_best_pass_sequence`), and final answers enclosed in `<answer>` tags. The environment executes these actions—simulating compilation tasks or tool invocations—and returns feedback in the form of updated state information and scalar rewards.

**Reward Design.** To guide the learning process, we define a composite reward function $R_{\text{final}} = w_f \cdot R_{\text{format}} + w_a \cdot R_{\text{answer}}$, where $w_f$ and $w_a$ control the relative importance of each component.

- *Format Reward*: $R_{\text{format}}$ encourages well-structured and logically coherent trajectories. A full reward is given if the interaction trace (including thoughts, tool usage, and answer) adheres to a predefined valid path (e.g., directly correct guess, or fallback via tool search); otherwise, no reward is assigned:

$$R_{format} = \begin{cases} 1.5, & \text{if the format is correct} \\ 0, & \text{if the format is incorrect} \end{cases} \quad (3)$$

- *Answer Reward*: $R_{\text{answer}} = \alpha \cdot \text{OverOrig}$ quantifies the effectiveness of the final sequence $\text{PassSeq}_{\text{answer}}$ in reducing the IR instruction count compared to the original unoptimized program $P_{\text{original}}$, where $\alpha$ is a scaling factor:

$$\text{OverOrig} = \frac{\text{Count}(P_{\text{original}}) - \text{Count}(\text{Apply}(P, \text{PassSeq}_{\text{answer}}))}{\text{Count}(P_{\text{original}})} \quad (4)$$

Unlike OverOz, which measures improvement over a compiler baseline, OverOrig provides a denser and more consistent reward signal, thereby facilitating stable RL training.

**Learning Algorithm.** We apply on-policy RL algorithms, specifically PPO [27], GRPO, and REINFORCE++ (RPP) [30], to update the LLM's policy. These algorithms are well-suited to LLMs, handling large action spaces and long-horizon credit assignment. Training is performed iteratively using collected trajectories, progressively refining the policy to generate higher-quality optimization sequences while maintaining training stability.

## 4 Experiments

This section presents the experimental evaluation of Compiler-R1. We begin by describing the common experimental setup, followed by three main experiments: (1) a performance comparison between Compiler-R1 and various baselines; (2) an analysis of factors affecting task success rate, which reflects effective environment interaction and highlights differences in sampling needs between interactive and non-interactive models; and (3) a case study investigating the impact of input feature representations.

### 4.1 Experimental Setup

**Datasets.** Experiments are conducted using an aggregated LLVM IR dataset. Training is performed on six CompilerGym datasets filtered to contain programs with fewer than 10k IR instructions. Evaluation is conducted on seven test suites: `blas`, `cbench`, `chstone`, `mibench`, `npb`, `opencv`, and `tensorflow`.

**Baselines.** We compare Compiler-R1 against classical compiler-tuning methods, including Open-Tuner [1], GA [13], TPE [4], RIO [7], CompTuner [31], BOCA [6], and AutoPhase [15] with PPO-LSTM and PPO-noLSTM variants.

**Optimization Space and Robustness.** All evaluated models operate within a fixed optimization space comprising 124 LLVM 10.0.0 `opt` passes and the `-Oz` preset (125 total actions). Robust evaluation protocols ensure reliability: Compiler-R1 defaults to `opt -Oz` if critical interaction failures occur (e.g., unparsable outputs or missing results). Non-interactive SFT models revert similarly upon invalid predictions. Compiler-R1 evaluates all successful interactions, whereas SFT models select the best valid sequence from $N$ inference attempts.

**Evaluation Metrics.** We define the following metrics:

- **Optimization Performance (**OverOz**%):** Average percentage IR instruction reduction relative to `opt -Oz` (Experiment 1).

- **Success Rate (%):** Proportion of test programs (335 total) for which models successfully execute the interaction protocol (e.g., correct tool invocation, output formatting). Applicable only to interactive models (Experiment 2).

- **Input Feature Impact (**OverOrig**):** Average IR instruction reduction relative to the original, unoptimized program to evaluate the impact of feature representations (Experiment 3).

**Models and Training.** All experiments were conducted on Intel Xeon Gold 6430 servers (128 cores, 1TB RAM) with NVIDIA H100 GPUs (4×80GB HBM3).

- **Compiler-R1 (Interactive Framework):** We use Qwen2.5-Instruct models (1.5B, 3B, 7B). Training involves 800 supervised fine-tuning (SFT) samples for protocol initialization, followed by reinforcement learning (GRPO, PPO, or RPP) on 19k interactive episodes, updating over 40 steps.

- **SFT-Only Baselines:** Non-interactive Qwen models (1.5B, 3B, 7B), fine-tuned to directly predict optimal pass sequences. Experiment 1 reports their best performance from $N = 40$ inference attempts per program; Experiment 2 examines their sensitivity by varying $N$ from 1 to 50.

**Additional Ablation Baselines (Experiment 2):**

- **SFT-only (Cold-start):** Qwen models (1.5B, 3B, 7B), trained only with the 800-sample interaction protocol initialization.
- **RL-only (No SFT):** Qwen-1.5B model trained exclusively via GRPO on the full RL dataset without protocol initialization.

## 4.2 Experiment 1: Optimization Performance Comparison

This experiment compares the average $\mathrm{OverOz}\%$ performance of Compiler-R1 variants with SFT-only models (which directly predict pass sequences) and traditional auto-tuning baselines. SFT-only models (SFT-Qwen-1.5B/3B/7B) are evaluated using $N = 40$ inference attempts per program, reporting the best-performing result per instance. Results from traditional methods and AutoPhase variants (PPO-LSTM and PPO-no-LSTM) are also included. Table 1 summarizes the findings.

Table 1: Comparison of Optimization Performance (Average $\mathrm{OverOz}\%$) and Time Cost. For SFT-Qwen models, $\mathrm{OverOz}\%$ results are from N=40 inference attempts.

| Method | blas (%) | cbench (%) | chstone (%) | mibench (%) | npb (%) | opencv (%) | tf (%) | Average (%) | Time (s) (avg/prog) |
|---|---|---|---|---|---|---|---|---|---|
| *Compiler-R1 (Interactive)* | | | | | | | | | |
| GRPO-1.5B (Ours) | 1.26 | 4.33 | 5.00 | 1.98 | 8.42 | 4.28 | 1.85 | 3.87 | 20 |
| GRPO-3B (Ours) | 3.14 | 3.77 | 2.97 | 3.84 | 15.42 | 3.77 | 2.96 | 5.12 | 22 |
| GRPO-7B (Ours) | **5.27** | 5.52 | **9.08** | **6.67** | 22.44 | **4.52** | **5.72** | **8.46** | 26 |
| PPO-1.5B (Ours) | 2.21 | 4.44 | 1.90 | 2.75 | 6.48 | 3.45 | 1.53 | 3.25 | 20 |
| RPP-1.5B (Ours) | 0.24 | 2.96 | 3.84 | 0.83 | 4.43 | 4.32 | 2.52 | 2.73 | 20 |
| *SFT-only (Direct Pass Prediction, N=40 attempts for* $\mathrm{OverOz}\%$*)* | | | | | | | | | |
| SFT-Qwen-1.5B (N=40) | 0.90 | 1.18 | 0.42 | 1.43 | 11.24 | 3.17 | 3.69 | 3.15 | 29 |
| SFT-Qwen-3B (N=40) | 0.78 | 0.62 | 0.88 | 0.68 | 7.00 | 2.62 | 1.15 | 1.96 | 40 |
| SFT-Qwen-7B (N=40) | 2.29 | 0.40 | 6.31 | 4.63 | 12.89 | 3.78 | 2.31 | 4.66 | 55 |
| *Traditional Autotuners* | | | | | | | | | |
| Opentuner | 1.60 | 1.99 | 6.46 | 3.33 | **26.19** | 1.76 | 1.29 | 6.09 | 200 |
| GA | -1.91 | 1.99 | 6.51 | 0.90 | 25.63 | 1.76 | 1.29 | 5.17 | 561 |
| TPE | -2.24 | 0.97 | 7.60 | 0.20 | 24.62 | 1.46 | 1.23 | 4.83 | 812 |
| RIO | -2.02 | 0.24 | 4.98 | 3.47 | 23.87 | 0.79 | 1.23 | 4.65 | 200 |
| CompTuner | -3.06 | -0.65 | 4.38 | -0.45 | 22.99 | 0.44 | 1.01 | 3.52 | 9803 |
| BOCA | -2.36 | -0.16 | 3.18 | -0.69 | 22.87 | 1.13 | 1.22 | 3.60 | 2760 |
| AutoPhase (PPO-LSTM) | -1.12 | **5.60** | 4.49 | 4.41 | -4.67 | -0.09 | 0.05 | 1.24 | 2.2 |
| AutoPhase (PPO-noLSTM) | -4.77 | -79.69 | -80.90 | -107.33 | -76.69 | -2.32 | -0.76 | -50.35 | 1.8 |

**Compiler-R1 achieves consistent and substantial improvements across all benchmarks**. Notably, GRPO-7B attains the highest average improvement of 8.46% $\mathrm{OverOz}$, significantly outperforming the strongest SFT-only baseline (SFT-Qwen-7B at 4.66%) despite using fewer inference attempts and less runtime. Lower-scale variants (GRPO-1.5B and GRPO-3B) also show competitive performance, further demonstrating the efficacy of RL-based interaction.

Compared to traditional autotuners, Compiler-R1 achieves comparable or better optimization quality with markedly lower runtime. GRPO-7B completes each program in  26 seconds, outperforming OpenTuner (200s), GA (561s), and CompTuner (9800s+). This efficiency makes Compiler-R1 viable for practical deployment.

Although AutoPhase variants benefit from extremely fast inference ( 2s), their optimization quality is notably limited. In contrast, **Compiler-R1 balances both performance and efficiency**: even the lightweight GRPO-1.5B achieves 3.87% in just 20 seconds, surpassing SFT-Qwen-1.5B (3.15% in 29s). These results affirm Compiler-R1's strong potential in real-world compiler optimization pipelines.

## 4.3 Experiment 2: Task Success Rate and Interaction Efficiency

This experiment evaluates the ability of different models to follow the predefined interaction protocol, quantified by the *task success rate* on 335 test programs. This metric reflects the agent's capacity to correctly execute the `thought–tool–answer` trajectory, serving as a proxy for effective environment interaction. We examine the effects of RL algorithms, model scale, training strategy (SFT-only, RL-only, SFT+RL), and repetition penalty. Additionally, we compare the interaction efficiency of

Compiler-R1 against SFT-only models under varying inference budgets $N$. Success rate results are summarized in Table 2, while Figure 3 visualizes the sensitivity of SFT-only models to $N$.

Table 2: Impact of Configuration and Repetition Penalty on Average Task Success Rate (%) on 335 test programs. Model group abbreviations: Comp.-R1 = Compiler-R1 (SFT Cold-start + RL), SFT (Abl.) = SFT-Qwen (Ablation), RL (Abl.) = RL-only GRPO (Ablation).

| | Comp.-R1 | | | | | SFT (Abl.) | | | RL (Abl.) |
| Rep. Pen. | GRPO 1.5B | PPO 1.5B | RPP 1.5B | GRPO 3B | GRPO 7B | SFT 1.5B | SFT 3B | SFT 7B | RL 1.5B |
|---|---|---|---|---|---|---|---|---|---|
| 1.05 | **83.36** | 65.97 | 71.04 | 45.56 | 51.92 | 26.98 | 4.93 | 36.85 | 22.55 |
| 1.10 | 79.69 | 60.95 | 60.45 | 42.61 | **96.71** | 41.08 | 2.72 | 9.50 | 8.92 |

As shown in Table 2, Compiler-R1 models consistently achieve high task success rates, with GRPO-1.5B reaching 83.36% and GRPO-7B attaining 96.71%. These results highlight the effectiveness of RL in enabling agents to reliably execute the thought-tool-answer trajectory mandated by the interaction protocol. This effective interaction—characterized by precise tool usage and sound decision-making—facilitates the robust optimization performance reported in Experiment 1 through a guided, single-trajectory search process.

Interestingly, GRPO-3B achieves a higher OverOz (5.12%) than GRPO-1.5B (3.87%) despite a much lower success rate (45.56%). This discrepancy illustrates that the success rate captures strict protocol compliance, whereas OverOz reflects end-to-end optimization effectiveness. GRPO-3B occasionally circumvents protocol checks (e.g., `instrcount` validation) by directly invoking `find_best_pass_sequence`. While this results in protocol violations and lower success scores, it can still yield high OverOz if effective sequences are discovered. In contrast, GRPO-1.5B more faithfully follows protocol, likely due to its smaller model capacity and greater adherence to SFT-initialized patterns. GRPO-7B strikes a favorable balance, combining protocol compliance and optimization quality through improved capacity and hyperparameter tuning.

SFT-only models, lacking interaction feedback, require extensive sampling to perform well. As shown in Figure 3, SFT-Qwen-7B achieves $-14.78\%$ OverOz at $N = 1$ but improves to 4.66% at $N = 40$, demonstrating their reliance on stochastic inference rather than adaptive refinement. Ablation results also confirm that neither SFT-only nor RL-only models are sufficient alone: only the two-stage SFT+RL pipeline consistently achieves high interaction reliability. Additionally, tuning repetition penalty proves crucial for success rate stability and overall interaction behavior.

In summary, Compiler-R1 exhibits strong interactive robustness across configurations and outperforms non-interactive models in both reliability and efficiency, validating the importance of protocol-aware training for compiler optimization agents.

Table 3: Comparative performance (Avg. Max OverOrig) of AutoPhase features versus raw LLVM IR as input representations for direct compiler pass sequence prediction using SFT with Qwen-1.5B models.

| Decoding / N | AutoPhase | LLVM IR |
|---|---|---|
| Greedy (N=1) | 0.4314 | 0.4321 |
| Sampled (N=1) | 0.3609 | 0.3921 |
| Sampled (N=10) | 0.4620 | 0.4643 |
| Sampled (N=20) | 0.4693 | 0.4713 |
| Sampled (N=30) | 0.4690 | 0.4690 |
| Sampled (N=40) | **0.4837** | 0.4795 |
| Sampled (N=50) | 0.4819 | **0.4810** |

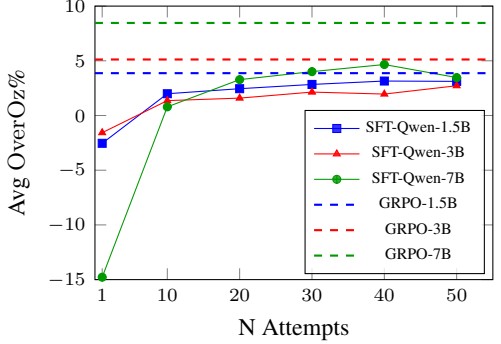

Figure 3: Average OverOz% of SFT-only models vs. N attempts. Dashed: GRPO Ref.

## 4.4 Experiment 3: Input Representation - AutoPhase vs. Raw LLVM IR

This experiment evaluates the impact of input representation on the effectiveness of compiler pass sequence prediction. Specifically, we compare AutoPhase features against raw LLVM IR as inputs to SFT models based on Qwen-1.5B, using $\mathrm{OverOrig}$ as the evaluation metric.

Two Qwen-1.5B models were trained on over 10,000 LLVM IR functions, with a held-out test set of 100 functions, each containing 1,800–3,000 IR tokens. One model was trained using 56-dimensional AutoPhase features; the other used tokenized raw LLVM IR. We evaluate each model under two decoding strategies:

- **Greedy Decoding** ($N = 1$): Deterministic decoding using argmax at each step.
- **Sampled Decoding** ($N = 1$–50): Nucleus sampling to generate up to 50 diverse candidate sequences per input, reporting the best $\mathrm{OverOrig}$ among them.

Table 3 presents the results, showing that AutoPhase features yield performance closely comparable to raw IR inputs.

**Discussion:** AutoPhase provides a compact and semantically rich input representation that performs comparably to raw LLVM IR in supervised pass sequence prediction. Under $N = 40$ sampling, the AutoPhase-based model achieves a $\mathrm{OverOrig}$ score of 0.4837, slightly higher than the 0.4795 attained using raw IR. This result suggests that AutoPhase captures sufficient structural and semantic information to guide effective optimization, while offering substantial advantages in terms of input length and interpretability.

One potential explanation is that AutoPhase abstracts away non-essential syntactic variations (e.g., variable names, function ordering), which can obscure semantic similarity in raw IR and lead to inconsistent model behavior. Moreover, performance generally improves with larger $N$, saturating around $N = 30$–50, which aligns with expectations under stochastic decoding. Notably, greedy decoding ($N = 1$) consistently outperforms single-shot sampling, highlighting the stability of deterministic inference for this task.

In addition, AutoPhase's fixed-size, 56-dimensional feature representation may contribute to generalization across programs of varying sizes. Because program size only influences the numerical values within this compact vector rather than the input dimensionality, the model is less affected by context length limitations that typically constrain raw-code-based methods. By relying on structural and proportional characteristics (e.g., the ratio of branch to memory instructions) rather than absolute code length, the model may learn scale-invariant patterns that transfer across different program complexities. While further empirical validation on larger program benchmarks is necessary, this feature-based representation appears to provide a practical foundation for scalable and potentially more robust learning.

In summary, AutoPhase serves as a strong, compact alternative to raw IR for LLM-based compiler tuning, particularly in scenarios constrained by context window length or requiring interpretable feature inputs.

## 4.5 Summary of Experimental Findings

Our experiments highlight Compiler-R1's effectiveness in compiler auto-tuning.

**First**, Compiler-R1—especially the GRPO-7B variant—achieves strong optimization results, with an average $\mathrm{OverOrig}$ gain of 8.46% and a task success rate of 96.71%. These results confirm the benefits of our two-stage interactive RL framework, which enables efficient environment interaction and high-quality pass sequence discovery. In contrast, SFT-only models require extensive sampling (e.g., $N = 40$) to reach comparable performance due to the lack of feedback-driven refinement.

**Second**, ablation studies demonstrate the necessity of two-stage training. The initial SFT phase is essential for learning interaction protocols that RL alone fails to acquire, while RL fine-tuning further improves efficiency and generalization.

**Finally**, AutoPhase features offer a compact and effective alternative to raw LLVM IR, delivering similar performance in sequence prediction with reduced input length and better robustness to irrelevant syntactic variations.

Together, these results position Compiler-R1's tool-augmented, two-stage LLM framework as a scalable and efficient solution for modern compiler optimization tasks.

# 5  Limitations and Future Work

This work examines the use of reinforcement learning to enhance LLM-based compiler auto-tuning. While the results show potential, several limitations remain and suggest directions for future study.

**Instruction Count as a Proxy for Performance**  We use the IR instruction count as the main optimization target. Although this is a common proxy for code size reduction, it does not always correlate with runtime improvements. Cache effects, instruction-level parallelism, and hardware-specific behavior can cause shorter instruction sequences to run slower. Future work could explore integrating direct metrics, such as execution time or cycle counts, into the reward function. This would increase measurement cost but better align the optimization objective with real performance.

**Semantic Information Loss in Program Representation**  We adopt the 56 statistical features from AutoPhase to represent programs. This compact form fits within the LLM's context but abstracts away structural and semantic details, such as control and data flow. This loss may limit the model's ability to capture complex optimization patterns. Future work could explore richer representations that preserve more semantics while keeping manageable size, such as graph encodings or embeddings.

**Reliance on External Tools**  Compiler-R1 relies on external tools to evaluate and search for optimization pass sequences. While this design is practical and aligns with how compilers are typically used, it also means that the model primarily learns to follow a procedural protocol rather than developing a deeper understanding of why a given sequence is effective. Future work could explore curriculum learning or related staged training strategies, where the agent is exposed to progressively more complex optimization tasks. This might help improve its general reasoning ability and reduce reliance on external evaluation tools.

**Cross-Version Compiler Compatibility**  The compiler ecosystem evolves over time, with optimization passes being added, removed, or modified between releases. The current dataset and training setup are tied to a fixed compiler version. When applied to a different version, the framework currently defaults to using the intersection of known passes, which may be conservative. Improving version adaptability remains an open problem. Possible directions include developing methods for automatic adaptation to unseen compiler versions, such as zero-shot or few-shot generalization to new passes, or incremental updates to datasets and reward functions to maintain compatibility with evolving compiler toolchains.

# 6  Conclusion

We present Compiler-R1, a tool-augmented framework for compiler auto-tuning with LLMs, addressing the lack of domain-specific training data and the limited adaptability of non-interactive methods. Compiler-R1 introduces a high-quality reasoning dataset combining chain-of-thought and tool-use paradigms, and adopts a two-stage training pipeline: SFT for initialization, followed by RL for outcome-driven policy optimization. Experiments show that Compiler-R1, particularly the GRPO-7B variant, achieves an average 8.46% IR instruction reduction and a 96.71% task success rate, outperforming SFT-only baselines and rivaling traditional autotuners with greater efficiency. Ablation studies highlight the importance of combining SFT and RL. Our results demonstrate the potential of LLM-based agents trained with reinforcement learning approaches in complex, interactive compiler optimization tasks.

Compiler-R1 showcases the profound potential of tool-augmented LLM agents trained via RL to revolutionize compiler auto-tuning, validating the effectiveness of an agentic LLM+RL paradigm for complex compiler auto-tuning tasks. Future research directions include leveraging LLMs to improve the interpretability of auto-tuning processes, further developing LLM-RL agents to explicitly understand and exploit synergies between compiler passes, and exploring the efficacy of interactive, turn-by-turn pass application dialogues that could simulate traditional RL auto-tuning paradigms.

# 7 Acknowledgements

This work is supported by the National Key R&D Program of China, Grant No.2022YFB4401402.

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

# Appendix

## A  AutoPhase Feature Set

As referenced in **Feature Extraction and Representation**, our framework utilizes the 56 statistical features from AutoPhase [15] to represent programs compactly for the LLM. These features capture various aspects of the program's static structure and instruction mix. Table 4 provides a complete list of these features.

Table 4: List of 56 AutoPhase Features Utilized.

| Index | Feature Description | Index | Feature Description |
|---|---|---|---|
| 0 | BBs: total phi args > 5 | 28 | Number of And insts |
| 1 | BBs: total phi args in [1,5] | 29 | BBs: instruction count in [15,500] |
| 2 | BBs: count with 1 predecessor | 30 | BBs: instruction count < 15 |
| 3 | BBs: count with 1 predecessor and 1 successor | 31 | Number of BitCast insts |
| 4 | BBs: count with 1 predecessor and 2 successors | 32 | Number of Br insts |
| 5 | BBs: count with 1 successor | 33 | Number of Call insts |
| 6 | BBs: count with 2 predecessors | 34 | Number of GetElementPtr insts |
| 7 | BBs: count with 2 predecessors and 1 successor | 35 | Number of ICmp insts |
| 8 | BBs: count with 2 predecessors and successors | 36 | Number of LShr insts |
| 9 | BBs: count with 2 successors | 37 | Number of Load insts |
| 10 | BBs: count with >2 predecessors | 38 | Number of Mul insts |
| 11 | BBs: Phi node count in range (0,3] per BB | 39 | Number of Or insts |
| 12 | BBs: count with more than 3 Phi nodes | 40 | Number of PHI insts |
| 13 | BBs: count with no Phi nodes | 41 | Number of Ret insts |
| 14 | Number of Phi-nodes at beginning of BB | 42 | Number of SExt insts |
| 15 | Number of branches | 43 | Number of Select insts |
| 16 | Number of calls that return an int | 44 | Number of Shl insts |
| 17 | Number of critical edges | 45 | Number of Store insts |
| 18 | Number of edges | 46 | Number of Sub insts |
| 19 | Occurrences of 32-bit integer constants | 47 | Number of Trunc insts |
| 20 | Occurrences of 64-bit integer constants | 48 | Number of Xor insts |
| 21 | Occurrences of constant 0 | 49 | Number of ZExt insts |
| 22 | Occurrences of constant 1 | 50 | Number of basic blocks |
| 23 | Number of unconditional branches | 51 | Number of instructions (all types) |
| 24 | Binary operations with a constant operand | 52 | Number of memory instructions |
| 25 | Number of AShr insts | 53 | Number of non-external functions |
| 26 | Number of Add insts | 54 | Total arguments to Phi nodes |
| 27 | Number of Alloca insts | 55 | Number of Unary operations |

## B  Synergy Pass Pair Algorithm Details

Algorithm 1 implements the synergy pass pair identification methodology described in Section 3.1. The procedure operates as follows:

## C  LLVM Optimization Passes

As detailed in Section 4.1, our Compiler-R1 framework and baseline models operate within an action space comprising 124 individual LLVM 10.0.0 `opt` transformation passes, augmented with the `-Oz`. This results in a total of 125 distinct actions available to the optimization agent. Table 5 enumerates these passes and the `-Oz` along with their corresponding indices used within our system. These flags can typically be obtained from the CompilerGym LLVM environment.

---

**Algorithm 1** Finding Synergy Pass Pairs for Program

---

**Require:** Program $P$, set of compiler transformation passes $O$
**Ensure:** Set of chained synergy pass pairs $S \leftarrow \emptyset$
 1: Compute $V_P$, the IR instruction count of the $P$
 2: **for** each optimization pass $B \in O$ **do**
 3:      Apply pass $B$ to $P$ and obtain the IR instruction count $V_{P_B}$
 4:      **if** $V_{P_B} < V_P$ **then**
 5:          **for** each optimization pass $A \in O$ **do**
 6:              Apply pass $A$ followed by pass $B$ to $P$ and obtain the IR instruction count $V_{P_{AB}}$
     compared to $O_0$
 7:              **if** $V_{P_{AB}} < V_{P_B}$ **then**
 8:                  $S \leftarrow S \cup \{(A, B)\}$
 9:              **end if**
10:          **end for**
11:      **end if**
12: **end for**
13: **return** $S$

---

Table 5: List of Utilized LLVM Compiler Passes and `-Oz` with Corresponding Indices.

| Index | Flag | Index | Flag | Index | Flag |
|---|---|---|---|---|---|
| 0 | add-discriminators | 42 | globalsplit | 84 | lower-expect |
| 1 | adce | 43 | guard-widening | 85 | lower-guard-intrinsic |
| 2 | aggressive-instcombine | 44 | hotcoldsplit | 86 | lowerinvoke |
| 3 | alignment-from-assumptions | 45 | ipconstprop | 87 | lower-matrix-intrinsics |
| 4 | always-inline | 46 | ipsccp | 88 | lowerswitch |
| 5 | argpromotion | 47 | indvars | 89 | lower-widenable-condition |
| 6 | attributor | 48 | irce | 90 | memcpyopt |
| 7 | barrier | 49 | infer-address-spaces | 91 | mergefunc |
| 8 | bdce | 50 | inferattrs | 92 | mergeicmps |
| 9 | break-crit-edges | 51 | inject-tli-mappings | 93 | mldst-motion |
| 10 | simplifycfg | 52 | instsimplify | 94 | sancov |
| 11 | callsite-splitting | 53 | instcombine | 95 | name-anon-globals |
| 12 | called-value-propagation | 54 | instnamer | 96 | nary-reassociate |
| 13 | canonicalize-aliases | 55 | jump-threading | 97 | newgvn |
| 14 | consthoist | 56 | lcssa | 98 | pgo-memop-opt |
| 15 | constmerge | 57 | licm | 99 | partial-inliner |
| 16 | constprop | 58 | libcalls-shrinkwrap | 100 | partially-inline-libcalls |
| 17 | coro-cleanup | 59 | load-store-vectorizer | 101 | post-inline-ee-instrument |
| 18 | coro-early | 60 | loop-data-prefetch | 102 | functionattrs |
| 19 | coro-elide | 61 | loop-deletion | 103 | mem2reg |
| 20 | coro-split | 62 | loop-distribute | 104 | prune-eh |
| 21 | correlated-propagation | 63 | loop-fusion | 105 | reassociate |
| 22 | cross-dso-cfi | 64 | loop-guard-widening | 106 | redundant-dbg-inst-elim |
| 23 | deadargelim | 65 | loop-idiom | 107 | rpo-functionattrs |
| 24 | dce | 66 | loop-instsimplify | 108 | rewrite-statepoints-for-gc |
| 25 | die | 67 | loop-interchange | 109 | sccp |
| 26 | dse | 68 | loop-load-elim | 110 | slp-vectorizer |
| 27 | reg2mem | 69 | loop-predication | 111 | sroa |
| 28 | div-rem-pairs | 70 | loop-reroll | 112 | scalarizer |
| 29 | early-cse-memssa | 71 | loop-rotate | 113 | separate-const-offset-from-gep |
| 30 | early-cse | 72 | loop-simplifycfg | 114 | simple-loop-unswitch |
| 31 | elim-avail-extern | 73 | loop-simplify | 115 | sink |
| 32 | ee-instrument | 74 | loop-sink | 116 | speculative-execution |
| 33 | flattencfg | 75 | loop-reduce | 117 | slsr |
| 34 | float2int | 76 | loop-unroll-and-jam | 118 | strip-dead-prototypes |
| 35 | forceattrs | 77 | loop-unroll | 119 | strip-debug-declare |
| 36 | inline | 78 | loop-unswitch | 120 | strip-nondebug |
| 37 | insert-gcov-profiling | 79 | loop-vectorize | 121 | strip |
| 38 | gvn-hoist | 80 | loop-versioning-licm | 122 | tailcallelim |
| 39 | gvn | 81 | loop-versioning | 123 | mergereturn |
| 40 | globaldce | 82 | loweratomic | 124 | `-Oz` |
| 41 | globalopt | 83 | lower-constant-intrinsics | | |

# D Illustrative SFT Answer Trajectories

## D.1 Answer 1: Initial Pass Sequence Outperforms Oz

```
<think>
[Initial Pass Sequence Analysis]
- Based on the provided autophase features, I'll try the following pass sequence:
["--dse","--correlated-propagation",...]
- First, I'll use the instrcount tool to check if this sequence provides improvement
    over -Oz optimization.
If improvement_over_oz > 0, I'll use this sequence as my answer.
If improvement_over_oz <= 0, I'll call find_best_pass_sequence tool to find a better seq.
</think>
<tool_call>
{"name":"instrcount","arguments":{"filepath":"example1.ll",
"optimization_flags":["--dse","--correlated-propagation",...]}}
</tool_call>
<tool_response>
{
  "status": "success",
  "improvement_over_oz": 0.06896551724137931
}
</tool_response>
<think>
[Result Analysis]
- The instrcount_tool reports an improvement_over_oz value of 0.06896551724137931.
- Since this value is positive, the pass sequence provides better optimization than -Oz.
- I'll use this pass sequence as my final answer.
</think>
<answer>
["--dse","--correlated-propagation",...]
</answer>
```

## D.2 Answer 2: Initial Sequence Underperforms, FBPS Finds Better Sequence

```
<think>
[Initial Pass Sequence Analysis]
- Based on the provided autophase features, I'll try the following pass sequence:
["--jump-threading","--dse","--mem2reg",...]
- First, I'll use the instrcount tool to check if this sequence provides improvement
    over -Oz optimization.
If improvement_over_oz > 0, I'll use this sequence as my answer.
If improvement_over_oz <= 0, I'll call find_best_pass_sequence tool to find a better seq.
</think>
<tool_call>
{"name":"instrcount","arguments":{"filepath":"example2.ll",
"optimization_flags":["--jump-threading","--dse","--mem2reg",...]}}
</tool_call>
<tool_response>
{
  "status": "success",
  "improvement_over_oz": -0.04265402843601896
}
</tool_response>
<think>
[Finding Better Pass Sequence]
- The previous sequence did not provide improvement over -Oz.
- Calling find_best_pass_sequence tool to find a better sequence.
</think>
<tool_call>
{"name":"find_best_pass_sequence","arguments":{"filepath":"example2.ll"}}
</tool_call>
<tool_response>
{
  "status": "success",
  "best_pass_sequence": ["--lower-expect","--reassociate",...],
  "improvement_percentage": 1.9
}
</tool_response>
<think>
[Final Decision - Found Improved Sequence]
- Found a pass sequence with positive improvement: 1.9.
- Using this sequence as the final answer.
</think>
<answer>
["--lower-expect","--reassociate",...]
</answer>
```

### D.3  Answer 3: Initial Sequence and FBPS Underperform, Fallback to Oz

```
<think>
[Initial Pass Sequence Analysis]
- Based on the provided autophase features, I'll try the following pass sequence:
["--mergereturn","--simplifycfg",...]
- First, I'll use the instrcount tool to check if this sequence provides improvement
     over -Oz optimization.
If improvement_over_oz > 0, I'll use this sequence as my answer.
If improvement_over_oz <= 0, I'll call find_best_pass_sequence tool to find a better seq.
</think>
<tool_call>
{"name":"instrcount","arguments":{"filepath":"example3.ll",
"optimization_flags":["--mergereturn","--simplifycfg",...]}}
</tool_call>
<tool_response>
{
  "status": "success",
  "improvement_over_oz": -0.02564102564102564
}
</tool_response>
<think>
[Finding Better Pass Sequence]
- The previous sequence did not provide improvement over -Oz.
- Calling find_best_pass_sequence tool to find a better sequence.
</think>
<tool_call>
{"name":"find_best_pass_sequence","arguments":{"filepath":"example3.ll"}}
</tool_call>
<tool_response>
{
  "status": "success",
  "best_pass_sequence": ["--simplifycfg", "--mem2reg",...],
  "improvement_percentage": 0.0
}
</tool_response>
<think>
[Final Decision - Fallback to -Oz]
- After attempting with find_best_pass_sequence tool, no sequence with positive
     improvement was found.
- Falling back to the standard -Oz optimization as the final answer.
</think>
<answer>
['-Oz']
</answer>
```

## E  Experiment 3 Prompt Templates

### E.1  AutoPhase Features Model

```
PROMPT:
Act as a compiler optimization expert finding an optimal pass sequence for LLVM IR,
aiming to reduce the total instruction count.
The LLVM IR code is represented by autophase features,
the initial autophase features are: {Autophase Features}
Initial instruction count: {initial_inst_count}

ANSWER TEMPLATE:
<answer>
[Recommended pass sequence goes here]
</answer>
```

### E.2  Raw LLVM IR Model

```
PROMPT:
Act as a compiler optimization expert finding an optimal pass sequence for LLVM IR,
aiming to reduce the total instruction count.
The LLVM IR code is: {ll_code}
Initial instruction count for this code: {initial_inst_count}

ANSWER TEMPLATE:
<answer>
[Recommended pass sequence goes here]
</answer>
```

