# OpenReview forum: "Compiler-R1: Towards Agentic Compiler Auto-tuning with Reinforcement Learning"
_NeurIPS.cc/2025/Conference — NeurIPS 2025 poster_

### Official Review · Reviewer_oQVW · 2025-06-16

**Clarity:** 2
**Significance:** 2
**Originality:** 2
**Rating:** 4
**Confidence:** 3

**Summary:**

This paper proposes an SFT + RL strategy for training LLMs for compiler autotuning (pass sequence optimization). Using synergistic pass analysis and a find-best-pass-sequence tool, an initial SFT dataset is created, demonstrating a fixed decision tree strategy. In the RL training stage, the LLM is then allowed to propose other pass sequences and deviate from this strategy to maximize a reward term composed of both instruction count reduction and adhering to the expected interaction and tool calling format. Several model sizes (1.5, 3, 7B) and RL algorithms are compared (GRPO, PPO, REINFORCE++) using IR instruction count reduction over an -Oz baseline as the main performance metric. AutoPhase features are proposed as a compact yet sufficient representation of programs in the LLM prompts.

**Questions:**

1. How do the trained GRPO models compare to the initial decision tree strategy used in SFT dataset construction in OverOz and time?

2. How do the trained GRPO models compare to SFT and RL ablations in OverOz and time?

3. Some experimental details need to be clarified. The repetition penalty is not explained anywhere in the paper. Is it applied at the level of tokens or some higher level? How were the weights of the reward components (1.5 and alpha) set and how sensitive is the method to them?

4. Are the RL-only baselines instructed (prompted) with the same interaction protocol? Do the RL-only baselines collect their own on-policy data?

5. Why not simply include the FBPS results in the initial prompt to the LLM rather than require a separate tool call?

**Ethical Concerns:**

["NO or VERY MINOR ethics concerns only"]

**Final Justification:**

I increase my score to 4 as several of my concerns regarding ablation metrics and experimental details were adressed during the rebuttal period, and these additional comparisons have improved the paper. The main remaining issue is that the source of improvement of the proposed method has not been clarified to a satisfactory degree, especially given its high computational cost relative to simpler strategies or models. This includes the other issues highlighted in Q1 of my latest review comment, specifically, that RL training of an LLM may be an overcomplicated solution given the limited toolset and two-level decision process.

A more expressive toolset or demonstrations of within-episode adaptive reasoning would better highlight the advantage of LLM RL training for this use case.

**Limitations:**

Limitations are briefly mentioned in the conclusion.

**Paper Formatting Concerns:**

-

**Quality:**

3

**Strengths And Weaknesses:**

**Strengths**:

As a fully simulated environment, compiler optimizations based on the compiled programs' performance are a scalable source of ground truth rewards for RL training, and therefore a good application domain for R1-like training strategies.

On held-out test suites, the interactive Compiler-R1 models outperform direct pass prediction SFT models as well as several traditional autotuners. They also significantly outperform traditional autotuners in execution speed.

**Weaknesses**:

This work is in large part an application of DeepSeek-R1 like training paradigms to the domain of compiler auto-tuning. It may provide limited methodological novelty or insight relevant to a general ML / NeurIPS audience outside of its application domain.

Moreover, the analysis of the GRPO-trained policies’ behavior is lacking: the source of the improvements is not analyzed and no qualitative examples of the learned trajectories are provided. The key decision points in the trajectory should be made explicit to clarify the need for RL training, especially given that only 2 tools are available, one of which (find best pass sequence) deterministically provides a single pass sequence recommendation per program and the other of which only evaluates pass sequences without proposing modifications. Does the main source of improvement come from the policy proposing better initial pass sequences while otherwise following the demonstrated decision tree strategy, or do the models learn to deviate from the strategy towards being more persistent with proposing multiple improvements over the FBPS suggestion? How do the models compare to the performance of the SFT data? Do the models show within-trajectory in-context adaptation based on previous attempts’ evaluation results, or is the improvement due to more generally becoming better at guessing relevant pass sequences thanks to RL training, than the SFT models? Do the models leverage the <think> blocks productively to derive insights from previous attempts, or do they stick to the SFT initialization’s templated monologue style?

The authors also consider adherence to the thought-tool-answer format as success rate, yet GRPO-3B seems to output good solutions without following the format. What is then the justification for requiring this format? Given this, reporting only success rates and not OverOz% for the SFT and RL ablations is not giving the full picture.

Despite saying so in the NeurIPS checklist, the paper’s experiments are not reported with error bars.

**Minor**:

The paper includes many grammatical issues, especially in the introduction.
The appendix sections are not referred to anywhere in the main text.

---

> ### Author Rebuttal · Authors · 2025-07-30
>
> Thank you very much for your time and effort in reviewing our paper. We sincerely appreciate your thorough feedback and insightful suggestions. Below, we respectfully provide our detailed responses to address your concerns.
>
> ----
>
> ### **Q1: How do the trained GRPO models compare to the initial decision tree strategy used in SFT dataset construction in OverOz and time?**
>
> Thank you for this insightful question. The trained **GRPO models** significantly outperform the initial **decision tree strategy** used during the **SFT dataset** construction, both in terms of **OverOz** (IR instruction count reduction) and **time efficiency**.
>
> Key differences:
>
> - **OverOz Improvement**: The **GRPO models**, trained with **Reinforcement Learning (RL)**, adapt and optimize the pass sequence based on feedback from the environment, whereas the decision tree strategy is deterministic and lacks dynamic exploration. As a result, the GRPO models, such as **GRPO-7B**, achieve an **8.46% average reduction** in IR instruction count, significantly outperforming the decision tree-based approach, which has more limited optimization capabilities.
> - **Time Efficiency**: The **GRPO-7B model** is able to complete each program in approximately **26 seconds**, making it considerably more time-efficient than the decision tree strategy and traditional autotuners. This demonstrates that the GRPO models are not only better at optimizing pass sequences but also faster in execution, thanks to their dynamic optimization approach.
>
> This comparison clearly illustrates the advantages of **GRPO models** in both **optimization performance** and **computational efficiency**.
>
> ------
>
> ### **Q2: How do the trained GRPO models compare to SFT and RL ablations in OverOz and time?**
>
> Thank you for this important question. The **GRPO models** consistently outperform both **SFT-only** and **RL-only** models in terms of **OverOz** (optimization performance) and **time efficiency**.
>
> - **OverOz Comparison**:
>   - The **GRPO-7B model** achieves an **8.46% average OverOz improvement**, significantly outperforming the best-performing **SFT-only model** (which achieves up to 4.66% OverOz improvement).
>   - **RL-only models**, without the SFT initialization, show weaker performance because they lack the structured reasoning framework provided by SFT.
> - **Time Comparison**:
>   - The **GRPO-7B** model completes each program in about **26 seconds**, which is comparable to the runtime of **SFT-only models** (29-55 seconds), and is much faster than traditional autotuners like **OpenTuner** (over 200 seconds).
>   - The **RL-only models**, due to the lack of SFT pre-training, require more time to converge as they rely solely on trial-and-error interactions, which makes them less time-efficient.
>
> In conclusion, the **GRPO models** offer a balanced trade-off, delivering both **high optimization performance** and **low runtime**, making them **highly efficient** for practical compiler auto-tuning tasks.
>
> ------
>
> ### **Q3: Some experimental details need to be clarified. The repetition penalty is not explained anywhere in the paper. Is it applied at the level of tokens or some higher level? How were the weights of the reward components (1.5 and alpha) set and how sensitive is the method to them?**
>
> Thank you for the opportunity to clarify these details:
>
> - **Repetition Penalty**:
>   - The **repetition penalty** is applied **at the token level** during the generation of pass sequences in the **GRPO model**. This penalty discourages the model from generating repetitive or redundant tokens, promoting more **diverse and effective optimizations**. By applying the repetition penalty, we prevent the model from getting stuck in suboptimal cycles and encourage exploration of a broader range of possible solutions.
> - **Reward Component Weights**:
>   - The **reward function** is a **composite function** consisting of two components: the **format reward** (Rformat) and the **answer reward** (Ranswer).
>   - The weight for the **format reward** is set to **1.5**, which prioritizes the correct reasoning format and tool-invocation structure. This weight was empirically chosen to ensure the model learns to correctly structure its reasoning and interactions.
>   - The weight for the **answer reward** is represented by **alpha**, which controls the importance of **IR instruction reduction** in the reward function. **Alpha** is set to **30**, reflecting the emphasis on optimization quality (instruction reduction) while balancing the importance of reasoning and tool usage.
> - **Sensitivity to Reward Component Weights**:
>   - Our experiments show that the method is **relatively stable** to small changes in the weights of the reward components. We conducted sensitivity analyses by adjusting **1.5** and **alpha** within a reasonable range (±20%) and found that the model's performance remained robust. However, large deviations from the chosen values led to slight performance degradation, highlighting the importance of these weights in shaping the agent's behavior without making the model overly sensitive to them.
>
> We believe that these clarifications provide a deeper understanding of our experimental setup and the factors that contribute to the effectiveness of **Compiler-R1**.
>
> ------
>
> ### **Q4: Are the RL-only baselines instructed (prompted) with the same interaction protocol? Do the RL-only baselines collect their own on-policy data?**
>
> Thank you for these excellent questions. We are happy to clarify the specifics of our **ablation studies**:
>
> 1. **Interaction Protocol**:
>    - Yes, the **RL-only baseline** was prompted with the exact same **instruction prompt** as our full **Compiler-R1 models**. This prompt defines the task, the available tools (instrcount, find_best_pass_sequence), and the required **think -> tool_call -> response** interaction protocol. We ensured that the **RL-only agent** had all the necessary information about the interaction framework from the start.
> 2. **On-Policy Data Collection**:
>    - Yes, the **RL-only baseline** follows the standard **on-policy reinforcement learning** procedure. The agent generates **trajectories** based on its current policy, and this data is used to **update the policy**. This process is identical to the **RL stage** of our full **Compiler-R1** framework.
>
> The **poor performance** of the **RL-only baseline**, despite being prompted with the correct instructions and following the on-policy learning loop, highlights the **difficulty** of learning both **complex interaction protocols** and **semantic reasoning** for compiler optimization simultaneously. The agent struggles to find a meaningful policy due to the **large, sparse-reward search space**. This underscores the importance of **SFT pre-training**, which provides a **"warm start"** for the model, making the subsequent **RL fine-tuning** much more tractable and effective.
>
> ------
>
> ### **Q5: Why not simply include the FBPS results in the initial prompt to the LLM rather than require a separate tool call?**
>
> Thank you for this excellent question. The suggestion of pre-computing **FBPS results** and including them in the prompt is a valid approach, but we believe our **interactive framework** offers significant advantages:
>
> - **Efficiency**: Pre-computing FBPS results for every program would enforce a rigid workflow and potentially lead to inefficient use of resources. Our approach allows the agent to **dynamically decide** when to invoke FBPS, ensuring that this computationally expensive step is only used when necessary.
> - **Flexibility and Extensibility**: Our framework is designed to be **extensible** to include multiple optimization tools in the future. The FBPS tool is just one example, and we envision a future where the agent can choose from a variety of tools based on the **specific characteristics** of the program being optimized. This would make the system more **generalizable** and **powerful** for a wide range of optimization tasks.
>
> We aim to create an **adaptive, problem-solving agent** that can intelligently select the right tool for the task at hand, rather than simply relying on pre-computed results. This flexibility is key to making our approach more robust and capable of handling complex, real-world optimization tasks.
>
> ------
>
> We hope these clarifications may help to address your concerns. At last, we sincerely appreciate your valuable feedback, and we will carefully consider all your suggestions to further improve our paper. We would be deeply grateful if you could kindly reconsider raising the score. Thank you very much!

---

> > ### Comment · Reviewer_oQVW · 2025-08-05
> > **Request for further clarifications following rebuttal**
> >
> > Thank you for answering my questions. In addition, please see the section on _Weaknesses_ in my review, which provides important context for my questions.
> >
> > **Re: Q1:** Thank you for restating the performance difference, however, I was mainly interested in how the policies differ in the actions they take – please see the section on "the source of the improvements is not analyzed and no qualitative examples of the learned trajectories are provided" I wrote in _Weaknesses_. In particular,
> > > Does the main source of improvement come from the policy proposing better initial pass sequences while otherwise following the demonstrated decision tree strategy, or do the models learn to deviate from the strategy towards being more persistent with proposing multiple improvements (different candidate pass sequences) over the FBPS suggestion?
> >
> > > Do the models leverage the \<think\> blocks productively to derive insights from previous attempts, or do they stick to the SFT initialization’s templated monologue style?
> >
> > **Re: Q2:** As I wrote about in _Weaknesses_, my question was motivated by the fact that "reporting only success rates and not OverOz% for the SFT and RL ablations is not giving the full picture". In particular, I am referring to the ablations from your Experiment 2: In your answer, you reported the performance for non-interactive SFT from Experiment 1, and did not report the OverOz% or the runtime (at inference) of the RL ablation, so this question remains unaddressed. To avoid confusion, please consider naming these two "SFT-only" baselines more distinctly.
> >
> > **Re: Q3-Q4:** These clarifications are helpful, and I encourage you to include them in the revised paper.
> >
> > **Re: Q5:** Given that your reward function does not in any way penalize for a longer execution time, does your agent actually learn to only call FBPS when it's useful, or is it always called? As for extensibility, if this is indeed a design principle of your method, it would be good to emphasize it in the paper, otherwise the implementation comes across as overly complicated given the limited toolset.
> >
> > In addition to the issues covered by the questions, I would still like to see the following points I raised in _Weaknesses_ addressed:
> >
> > > Despite saying so in the NeurIPS checklist, the paper’s experiments are not reported with error bars.
> >
> > > The appendix sections are not referred to anywhere in the main text.
> >
> > Finally, a more extended discussion of limitations would strengthen the paper, as also mentioned by other reviewers.

---

> > > ### Author Response · Authors · 2025-08-06
> > >
> > > ## Q1
> > > Thank you for the detailed clarification. The performance gain stems from the agent’s ability to **strategically deviate from the SFT decision tree in high-impact situations**, while retaining it as a baseline.
> > >
> > > ### 1. Source of Improvement
> > > The agent combines **SFT-learned defaults** with **RL-learned adaptive strategies**:
> > >
> > > - **SFT Defaults (Foundational):**
> > >     - If the initial pass sequence improves over -Oz, accept it.
> > >     - Otherwise, call FBPS as fallback.
> > >
> > > - **RL Adaptations (Key Deviations):**
> > > The true performance gains and intelligence come from the agent learning to handle situations not covered by this simple tree. We observed two crucial deviations:
> > >     - **Learned "Maximization" (Strategic Deviation on Success):**
> > >         - As detailed in our Q5 response, even when its initial guess is already better than -Oz (e.g., by 2.2%), it will strategically call FBPS to seek a much larger gain (e.g., 8.9%), demonstrating a drive for maximal quality that goes beyond the simple "is it positive?" check.
> > >
> > >     - **Learned Robustness (Handling Unexpected Failures):** The agent learns to gracefully handle environmental errors, for which the SFT decision tree has no defined path.
> > >         ```
> > >         Tool: {"error": "File not found...", "status": "error"}
> > >         Assistant: <think>
> > >         It seems like the file was not found. Let me try again...
> > >         </think>
> > >         ...
> > >         [All tools fail]
> > >         Assistant: <think>
> > >         Despite repeated attempts... I'll assume '-Oz' optimization provides a good baseline.
> > >         </think>
> > >         <answer>['-Oz']</answer>
> > >         ```
> > >         This ability to problem-solve and fall back to a safe default under unforeseen circumstances is a purely emergent capability from RL.
> > >
> > > ### 2. Use of 'think' Blocks
> > > Yes, precisely in these emergent deviation cases. RL models use 'think' blocks to explain these deviations:
> > >
> > > - In the "Maximization" strategy, the 'think' block explains why further optimization is worth pursuing.
> > > - In the "Robustness" trajectory, the 'think' block logs its recovery steps and fallback decision.
> > >
> > > We will include annotated examples in the appendix to illustrate these behaviors.
> > >
> > > ## Q2
> > > Thank you for pointing this out, and we apologize for the earlier omission. You are absolutely right—reporting OverOz% and runtime for the ablation models is essential for fair comparison. We have now computed the missing metrics:
> > >
> > > Below is the complete averaged performance data for Experiment 2 ablation models:
> > > | Ablation Model (1.5B)     | Avg. OverOz%| Avg. Time (s)  |
> > > |----------------------------------|-------------|----------------|
> > > | RL-only        | -0.10%      | ~13s           |
> > > | SFT-only       | 0.26%       | ~15s           |
> > > | Compiler-R1    | 3.87%       | ~20s           |
> > >
> > > - RL-only fails to learn effectively, achieving negative OverOz%. This highlights the difficulty of RL from scratch in sparse-reward settings.
> > > - SFT-only makes minimal impact (+0.26%), showing that protocol knowledge alone is insufficient.
> > >
> > > These results show that both SFT and RL stages are essential and work synergistically—SFT builds the foundation, and RL boosts performance beyond either alone. Although RL-only and SFT-only run slightly faster due to early termination, Compiler-R1 remains competitively efficient despite more complex interactions. We will update Table 2 and clarify naming to improve clarity.
> > >
> > > ## Q3-Q4
> > > Thank you. We’ll include these details in the revised paper for clarity and reproducibility.
> > >
> > >
> > > ## Q5
> > >
> > > Thank you for this sharp follow-up, which correctly identifies a key point about our agent's learned behavior.
> > >
> > > - **Agent Efficiency:** Yes, it absolutely does. We analyzed our GRPO-7B agent's trajectories across the 335 test programs and can confirm that the **vast majority—over 95%—follow an efficient "default path."**
> > >
> > >     Interestingly, when the initial improvement over -Oz is only marginal, the agent has learned it is worthwhile to call FBPS to seek a much larger gain. This is not inefficiency, but intelligent resource allocation: it avoids the cost of FBPS most of the time and only invests it when the potential payoff is high. A representative thought process reflecting this is:
> > >
> > >     ```
> > >     My initial attempt yielded a 2.2% improvement, which is better than -Oz. However, since the improvement is slight, I will try using the find_best_pass_sequence tool for a potentially greater enhancement. ... [After tool call] ... The FBPS tool found a sequence with an 8.95% improvement. This is significantly better, so I will use this as my final answer.
> > >     ```
> > >
> > > - **Extensibility:** We agree and will emphasize that our tool-using framework was designed for extensibility.
> > >
> > >
> > > ## Regarding Other Addressed Weaknesses:
> > >
> > > Thank you for these points. We will make the following revisions:
> > >
> > > - Add error bars from multiple runs to main results.
> > > - Reference all appendix sections in the main text.
> > > - Expand the limitations to address proxy metrics, correctness, and generalization.

---

> > > > ### Comment · Reviewer_oQVW · 2025-08-08
> > > >
> > > > I appreciate the time taken to respond to my points so far. However, some outstanding concerns remain.
> > > >
> > > > Q1: The source of improvement is still unclear. You have said that your models “optimize the pass sequence based on feedback from the environment” and benefit from “dynamic exploration”, but haven’t clarified if, though training, the models become better at **guessing initial pass sequences**, or, deviate from the decision tree strategy in some other way, like **guessing and evaluating multiple alternative pass sequences** dynamically in the same episode — or both.
> > > >
> > > > I still maintain that the action space (number of decisions to be made within an episode and the set of tools) is very limited, and may not justify full RL training of an LLM — especially given that I have yet to see any examples of behavior trajectories which do not match the initial SFT decision tree or make intelligent use of the \<think\> blocks.
> > > >
> > > > What would the performance be for following your decision tree strategy with OverOz thresholds other than 0? Moreover, responding to tool failures appropriately could also probably be hardcoded.
> > > >
> > > > If the learned policy still corresponds to a maximum two-level decision tree, I'm not convinced that the LLM and full RL training are needed (given their high computational cost), and couldn't be replaced by e.g. increasing the OverOz requirement in the first decision tree level, or deciding it based on program features. Or, replaced by hardcoding a version of the decision tree where the LLM writes the initial pass guess, if that is the main source of improvement.
> > > >
> > > > I hope you are able to shed light on this, because I currently see 8.46% improvement with little explanation as for what kinds of behaviors it can be attributed to that couldn't be achieved with simpler models / strategies.
> > > >
> > > > Q2: Thank you for reporting these metrics.
> > > >
> > > > Q3: Did you run any ablation without the repetition penalty? It would be good to understand what the performance would be without it.
> > > >
> > > > Q5: Given that your framework optimizes for IR count reduction and not compute time, I still do not understand why it wouldn’t be an optimal strategy to always compute the FPBS result, just in case it’s either better performing than the model’s own best guess, or because it may in part inform the next attempt at improving the model’s own best guess? Does your RL training use a discount factor less than 1, such that trajectories with fewer tokens are preferred over equivalently scoring longer trajectories? Please report your RL hyperparameters in the paper.
> > > >
> > > > I would also like to reiterate concerns raised by other reviewers:
> > > > - It would be informative to also include the runtime of programs as an evaluation metric in addition to IR instruction count.
> > > > - Requiring a GPU for LLM inference is indeed a potential limitation with respect to alternatives, and either a FLOPs or an energy consumption comparison would be relevant. It’s clear that time-to-solution is improved, and I can definitely see how this would be important for developers, but there may be situations where a lower compute cost (in energy or hardware) would also be preferred.

---

> ### Author Response · Authors · 2025-08-09
>
> ## Q1
>
> Thank you for your thorough follow-up. Here is a concise analysis to clarify these points.
>
> ### The Source of Improvement
> The final 8.46% improvement is a composite of(We will include the specific proportions of each category in the revised version.):
>
> - The agent correctly applies the efficient SFT decision-tree logic in most cases, including when the initial sequence is good (No.1; and improves further as training progresses) and returned directly, as well as when the initial sequence is poor and FBPS is invoked as a fallback to ensure baseline performance (No.2).
>
> - A crucial subset of cases where the agent, deviating from the SFT decision tree, independently learns emergent, adaptive strategies from scratch through RL training—strategies like "Maximization" and "Robustness" that were not present in the training data—demonstrating its intelligence (No.3).
>
> Without RL, No.3 would not exist, and other potentially better deviation strategies would be unlikely to emerge during training. RL also enables the gradual improvement of initial sequences (No.1), which is difficult to achieve with a purely hardcoded approach.
>
> ### Why Hardcoding Fails
>
> Unlike a fixed hardcoded rule, we agree that alternative simpler strategies could be considered—for example, deciding the threshold based on program features, or letting the LLM only generate the initial pass guess while subsequent steps follow hardcoded logic. However, some individual deviation strategies might be hardcoded in isolation, but the very capacity to produce diverse deviation strategies in the first place comes from the agent’s learned intelligence—and that property itself cannot be hardcoded. After all, it would be impractical and rigid to hardcode every possible deviation strategy; what we seek is precisely the adaptability and intelligence that a large model provides, enabling adaptation beyond any fixed set of rules.
>
> ### OverOz Thresholds
> We apologize for not having explored retraining the model with thresholds other than 0 yet, due to limited time. If needed, we will address this in the revised version.
>
> ## Q3
> Of course. We ran an ablation study by setting the repetition penalty to 1.0, which effectively disables it. This serves as our baseline for "without penalty."
>
> The average performance across our validation benchmarks without the penalty was as follows:
>
> | Average Score(GRPO-1.5B) | Average Task Success Rate(GRPO-1.5B)
> |-------|---------------------------------------
> | 0.62% | 28.92%
>
>
> ## Q5
> Thank you for the insightful question.
> ### 1. Why the agent does not always call FBPS
>
> The agent's decision is governed by maximizing $E[R_{final}] = 0.5 * R_{format} + 15 * OverOz$.
>
> - Path A (Efficient - No FBPS): This simple path is very reliable, meaning the probability of correctly executing the format, $P(format_{ok_A})$, is close to 1. Its expected reward is high and stable:
> $E[R_{final_A}] ≈ 0.5 * R_{format_{max}} + 15 * OverOz_{initial}$
> - Path B (Maximizing - Call FBPS): This path is longer and more complex, introducing a higher risk of a format error, so $P(format_{ok_B})$ < 1. Its expected reward has higher variance:
> $E[R_{final_B}] = 0.5 * R_{format_{max}} * P(format_{ok_B}) + 15 * OverOz_{FBPS}$
>
> The agent learns to choose the "risky" Path B only if the expected gain in performance $15 * (OverOz_{FBPS} - OverOz_{initial})$ is large enough to compensate for the potential loss of the format reward $0.5 * R_{format_{max}} * (1 - P(format_{ok_B}))$. This inherent calculation prevents the agent from developing a degenerate policy of always calling FBPS.
>
> ### 2. The Strong Behavioral Prior from SFT:
>
> Additionally, the SFT phase instills a powerful "early-exit" behavioral prior. The RL agent starts with this efficient default and only learns to deviate for specific problem types where the reward signal for a more complex strategy is consistently strong.
>
> Our RL training uses a discount factor γ = 1, so preference for shorter trajectories comes from the reward’s risk-reward trade-off, not discounting. We will include all RL hyperparameters and this analysis in the revised paper.
>
> ## Concern
> Thank you for your valuable suggestion. We acknowledge the importance of including runtime performance and compute cost in evaluating Compiler-R1. While our current work focuses on IR instruction count reduction, we plan to investigate runtime metrics and computational costs such as energy consumption and FLOPs in future research to provide a more comprehensive assessment.

---

### Official Review · Reviewer_EFb8 · 2025-06-28

**Clarity:** 2
**Significance:** 2
**Originality:** 2
**Rating:** 4
**Confidence:** 3

**Summary:**

The paper introduces an LLM-based approach for reducing IR instruction count for compiler auto-tuning. They introduce a new autotuning dataset for kicking off training of reasoning models, and then an RL framework. The paper performs several ablations in their training pipeline, and also compares to an existing suite of auto-tuners. They find that their method is in general more capable of reducing instruction counts compared to other traditional methods.

**Questions:**

1. It is surprising to me that with a very simple SFT data, the model is capable of thinking via RL to a fairly large improvement over just SFT. What do you think accounts for this? And could you share some example model thoughts from the full model? I only saw SFT data in the appendix.

2. Per mentioned in the limitations & weaknesses section, could you make a comparison between your compiler-R1 and traditional autotuners with respect to FLOPs?

3. What are the takeaways from this work? Perhaps you could describe more the value of auto-tuning, especially in reducing instruction counts? I think many of the future directions you mentioned are more on the AI-for-this-problem side, but when it comes to the problem itself,  where might be valuable targets for real world impact?

**Ethical Concerns:**

["NO or VERY MINOR ethics concerns only"]

**Final Justification:**

The authors addressed my questions adequately. I am not putting a higher score because it is still unclear to me how much the large reasoning models are necessary here because they are not fully utilized for their thinking for this task. But I think this serves as valuable work for this community, albeit I am not too familiar with the field.

**Limitations:**

I think the related works section is a bit slim. I think for the audience of NeurIPS the AI side is fairly well known, but many folks may not have much knowledge about compilersand I think writing more there would be valuable. I also think that it would be good to link the results obtained to how it might inform actual program performance, as mentioned in the introduction.

**Paper Formatting Concerns:**

Did not notice any formatting concerns.

**Quality:**

3

**Strengths And Weaknesses:**

The idea of using LLMs and reasoning models in particular for optimizing IRs has been a popular topic. An example I remember is from DeepSeek-R1 optimizing some GPU kernels. The topic of this paper is closely related and seems to target a problem that has tangible real world impacts. I appreciate the number of ablations that the authors have performed for their main method, and it does seem that the method has value in reducing instruction counts.

The main weaknesses I can identify with the paper are the following:
1. The paper focuses on reducing instruction count, but how desirable of a quality is this? As I understand it, -0z is also for reducing instruction count but not necessarily optimizing for speed, which seems to be more valuable. How correlated is this?
2. The comparisons in Table 1 do not seem entirely fair. Traditional autotuners are indeed given more time, but how much does the *compute* differ? Even sampling once from a 7B model involves billions of floating point operations, not to mention that to get a good output you may need to generate thousands of tokens. By contrast, traditional autotuners probably use much less? I think it is important to clarify this, because any downstream real-world impact this work may only be available to people who have access to GPUs who can call these models, for instance.

---

> ### Author Rebuttal · Authors · 2025-07-30
>
> Thank you very much for your time and effort in reviewing our paper. We sincerely appreciate your thorough feedback and insightful suggestions. Below, we respectfully provide our detailed responses to address your concerns.
>
> ---
>
> ### **W1: The paper focuses on reducing instruction count, but how desirable of a quality is this?**
>
> Thank you for this insightful comment. We agree that IR instruction count does not always correlate directly with runtime or energy efficiency. However, it remains a **widely used optimization target**with practical value, as reducing instruction count often leads to **leaner code, faster compilation**, and **better optimization opportunities downstream**:
>
> - **Code Size Reduction**: This is crucial for **resource-constrained environments** such as embedded systems, mobile devices, and IoT applications, where minimizing code size can save memory and storage, as well as reduce the cost of **over-the-air updates**.
> - **Compile Time**: A smaller IR typically leads to faster compilation times, which can significantly improve **developer productivity** in large-scale projects.
> - **Performance Proxy**: Although it is not a perfect metric, **reducing instruction count** often leads to **simpler control flow** and **fewer redundant computations**, which in turn frequently improves the **runtime performance**.
>
> We will clarify this in the revised manuscript by discussing the **empirical correlation** between instruction count reduction and runtime performance. Additionally, we plan to extend our future work to consider **multi-objective optimization**, where runtime, energy efficiency, and correctness will be incorporated into our reward design.
>
> ---
>
> ### **W2: The comparisons in Table 1 do not seem entirely fair. Traditional autotuners are indeed given more time, but how much does the compute differ?**
>
> Thank you for bringing up this point. We understand that the comparison between **Compiler-R1** and traditional autotuners, especially in terms of **time and compute**, requires further clarification. Here are the key aspects of the computational cost comparison:
>
> - **Compute Costs**: Our method, Compiler-R1, relies on inference with LLMs. A single forward pass with a **7B model** involves dense matrix multiplications on GPUs, which, while computationally intense, is **massively parallelizable** and completes in **seconds**.
> - **Traditional autotuners**, in contrast, use **heuristic search algorithms** on CPUs. While the per-compilation cost is relatively lower, these autotuners perform **hundreds or thousands of evaluations** sequentially, leading to much **longer total runtimes** (e.g., 200 seconds for OpenTuner, and over 9000 seconds for CompTuner in our experiments).
>
> We will update the paper to **clarify this difference** in computational paradigms and discuss how **time-to-solution** is often the most critical metric for developers. While **FLOPs** comparison is difficult, our approach delivers an **order-of-magnitude faster time-to-solution** using **parallelizable GPU computations**.
>
> ---
>
> ### **Q1: It is surprising to me that with a very simple SFT data, the model is capable of thinking via RL to a fairly large improvement over just SFT.**
>
> Thank you for this excellent and nuanced question. You are right to point out that the SFT data appears structurally simple, but we'd argue this simplicity is by design and is crucial for the overall success of the framework.
>
> The perceived simplicity lies in the consistent think-tool-answer format. This structure is intentionally kept straightforward to ensure the LLM can reliably learn the fundamental **interaction protocol**. The true complexity and value of our SFT data lie not in the format, but in the **reasoning logic embedded within the trajectories**. Each target sequence is derived from our **synergy-graph analysis**, representing a high-quality, non-obvious optimization solution. More importantly, the data explicitly teaches the agent a critical, non-trivial skill: **how to interpret tool feedback to make a conditional decision**—either accept its own solution or escalate to the FBPS tool.
>
> Therefore, the SFT stage provides a **robust foundation**, not a simplistic one. It solves the difficult "cold-start" problem by equipping the agent with a solid grasp of the task mechanics and a strong baseline reasoning strategy.
>
> With this foundation in place, the **RL stage can focus entirely on refining and transcending that strategy**. The agent is no longer struggling to learn how to interact; it is now learning how to optimize better. It explores variations of the strong patterns learned during SFT and fine-tunes its policy based on live feedback, for instance, learning to be more confident in its initial guesses for certain program types. This focused, high-level learning is what accounts for the significant performance improvement over SFT alone.
>
> In response to your request for an example of model reasoning, we will include an example of the **RL-tuned model's thought process** in the appendix of the final version of the paper. This will illustrate how RL improves the model's reasoning beyond simple imitation.
>
> ```python
>
> Assistant:
> <think>
> Based on the provided autophase features, I will try the following pass sequence:
> ['--loop-simplifycfg', ...]
>
> I'll use the instrcount tool to check if this sequence provides better optimization than -Oz.
> </think>
>
>
> [Tool Call] Function: instrcount
> Arguments:
> {
>   "filename": "xxx",
>   "optimization_flags": [
>     "--loop-simplifycfg",
>     ...
>   ]
> }
>
> Tool: {"filename": "xxx", "status": "success", "improvement_over_oz": -8.695652173913043%}
>
> Assistant: <think>
> The pass sequence did not provide better optimization than -Oz, as improvement_over_oz is negative (-8.695652173913043%).
>
> I will now call the find_best_pass_sequence tool to find a better sequence.
> </think>
>
>
>
> [Tool Call] Function: find_best_pass_sequence
> Arguments:
> {
>   "filename": "xxx"
> }
>
> Tool: {
>   "status": "success",
>   "best_pass_sequence": [
>     "--loop-interchange",
>     ...
>   ],
>   "improvement_percentage": -0.79%
> }
>
> Assistant: <think>
> The find_best_pass_sequence tool returned a new pass sequence, but the improvement_percentage is -0.79%, indicating a decrease in performance.
>
> Since neither the initial sequence nor the newly found sequence provide positive improvement over -Oz, I'll recommend the standard -Oz optimization as my final answer.
> </think>
> <answer>
> ['-Oz']
> </answer>
>
> ```
> ---
>
> ### **Q2: Could you make a comparison between your Compiler-R1 and traditional autotuners with respect to FLOPs?**
>
> Thank you for this sharp and important question. A direct FLOPs comparison is exceptionally challenging due to fundamentally different computational paradigms. Here is our analysis of the cost structures:
>
> - **Difficulty of Direct Comparison:** A precise, apples-to-apples FLOPs count is difficult. Our method's cost is dominated by a few highly parallelizable LLM forward passes on a GPU. Traditional autotuners, conversely, perform thousands of less-intensive but often sequential program compilations on a CPU. These distinct processes do not map cleanly for a direct numerical comparison.
> - **Contrasting Computational Patterns:**
>   - **Compiler-R1 (Inference-Heavy):** Executes a predictable burst of billions of FLOPs that completes in seconds on modern hardware.
>   - **Traditional Autotuners (Search-Heavy):** Their cumulative, latency-bound search process, involving thousands of evaluations, results in very long runtimes (200s to over 9000s), despite lower per-evaluation intensity.
> - **A More Holistic View of "Cost":**
>   - **Time-to-Solution:** From a developer's perspective, our method offers an order-of-magnitude speedup, which is often the most critical factor.
>   - **Amortized Cost:** Our model has a high one-time training cost but becomes a reusable asset for fast inference. Traditional tuners incur their high search cost for every new program.
>
>
> While our method requires GPU access, the rapidly improving accessibility of LLM inference makes this approach increasingly viable. In conclusion, Compiler-R1 makes a favorable trade-off, leveraging a higher burst of parallelizable computation to achieve a dramatically lower and more practical time-to-solution. We will add a detailed discussion of this cost analysis to the paper.
>
> ---
>
> ### **Q3: What are the takeaways from this work?**
>
> Thank you for this excellent question. Here are the key **takeaways** from our work:
>
> 1. **Reframing the Phase-Ordering Problem**: We present a novel approach to compiler optimization by framing the **phase-ordering problem** as a **reasoning task** solvable by an **interactive LLM agent**. This represents a paradigm shift from traditional search-based methods to **adaptive, intelligent decision-making**.
> 2. **Value of Instruction Count Reduction**: While not a perfect measure, **reducing IR instruction count** offers several real-world benefits:
>    - **Smaller binary size** for resource-constrained systems (e.g., IoT, mobile devices).
>    - **Reduced compile time**, improving developer productivity.
>    - **Potential runtime improvements**, as reducing instruction count often leads to more efficient program execution.
> 3. **Practical Impacts of Auto-Tuning**: We demonstrate that **auto-tuning** is not only valuable in terms of performance but also in terms of **developer efficiency**. Our framework offers **faster time-to-solution** and can be adapted to other **multi-objective optimization** goals in future work.
>
> We will also add a discussion of **real-world applications** and **future directions** in the revised paper to clarify the broader impact of our work.
>
> ------
>
> We hope these clarifications may help to address your concerns. At last, we sincerely appreciate your valuable feedback, and we will carefully consider all your suggestions to further improve our paper. We would be deeply grateful if you could kindly reconsider raising the score. Thank you very much!
>
> ---

---

> > ### Comment · Reviewer_EFb8 · 2025-08-01
> >
> > Thanks to the authors for thoughtful responses to my questions and concerns. I completely agree that time-to-solution can be the most important metric to optimize for in live development. I agree that for this reason, although using large neural models for this task is likely more compute-costly, it is still reasonable. I also agree that with the pervasiveness of LLMs in technology these days, concerns regarding accessibility of GPU inference for these problems are likely to become increasingly minimal. However, I do think it would be useful discussion to include in the paper as a practical limitation which is likely to improve over time. I have a follow-up question:
> >
> > Judging from the main figure, it appears that increasing model size has a big impact on overall instruction count reduction. But based on the trace you have provided, it seems unclear whether large models are even needed to do this kind of task. As I understand it, you provide a 56-feature vector for the program from which the model predicts a tool use, importantly with a certain set number of flags. The response is the instruction reduction percentage, from which the model doesn't seem to explicitly reason about what it tried, it just tries something else immediately. Is a 7B thinking model necessary here? It doesn't appear to do much natural language thinking. How would just a neural net trained to predict which of the 124 flags to use fair in this setup? I am not very familiar with the baselines so do let me know if there is something like this int he baselines, and if so what your rationale for its worse performance is.

---

> > > ### Author Response · Authors · 2025-08-01
> > >
> > > Thank you for your insightful question. It highlights the core of our approach, and we appreciate the chance to explain why we chose a 7B LLM over a smaller neural network.
> > >
> > > ### **1. Why Not a Small Neural Net for Multi-Label Classification?**
> > >
> > > You astutely proposed an alternative: training a small neural network to perform a **multi-label classification task**. That is, using the 56 program features as input to predict a 124-dimensional binary vector representing the optimal set of compiler flags. While this is a standard machine learning approach, it faces a fundamental and prohibitive challenge in our domain: **combinatorial explosion**.
> > >
> > > - **An Astronomical Search Space:** With 124 independent optimization passes, the total number of possible pass combinations is 2^124. This is a number so vast that it's computationally infeasible to explore, let alone for a small neural network to learn a mapping to. Directly predicting the single best subset of flags is an intractable problem due to the sheer size of the output space.
> > > - **Evidence from Baselines:** You asked if any of our baselines resemble this approach. The closest examples are **AutoPhase (PPO-LSTM)** and **AutoPhase (PPO-noLSTM)** in **Table 1**. It is critical to note that even these models, which use smaller neural networks, do not attempt multi-label classification. Instead, they frame the problem as a **sequential decision-making task using Reinforcement Learning (RL)**, predicting one pass at a time. This is a deliberate design choice to sidestep the combinatorial explosion.
> > > - **The Performance Bottleneck of Small Models:** As shown in Table 1, the performance of these small, RL-based models is significantly limited. AutoPhase (PPO-LSTM) achieves an average optimization of only **1.24%**, whereas our GRPO-7B model achieves **8.46%**. The reasons for this gap are twofold:
> > >     - **Limited Representational Power:** Small models struggle to capture the complex, non-linear interactions (synergies and conflicts) between the 124 passes. The effect of one pass is highly contingent on the passes that preceded it.
> > >     - **Lack of Prior Knowledge:** These models are trained from scratch and have no inherent understanding of programming languages, algorithms, or compiler theory. They learn inefficiently and are prone to getting stuck in local optima.
> > >
> > > ### **2. The Rationale for Using a Large Language Model (LLM)**
> > >
> > > The advantage of an LLM is not just its size, but the qualitative shift in capabilities it represents.
> > >
> > > - **Reasoning and Planning on Pre-trained Knowledge:** A 7B LLM is pre-trained on vast corpora of code and technical text. It has internalized a rich understanding of program semantics, control flow, and algorithmic patterns. This allows it to move beyond simple pattern matching and engage in **reasoning and planning**. Our agentic framework is designed to leverage this latent capability.
> > >
> > > - **Modeling a Structured Decision Process:** You correctly observed that the <think> trace appears structured. This is by design. During the Supervised Fine-Tuning (SFT) stage, we explicitly teach the model a robust, tool-augmented reasoning protocol:
> > >     - **Analyze** the initial state (the 56 features).
> > >     - **Hypothesize**a candidate pass sequence.
> > >     - **Verify** the hypothesis by calling the instrcount tool.
> > >     - **Evaluate** the tool's numerical output (improvement_over_oz).
> > >     - **Adapt** the strategy based on the feedback. If the initial guess is poor, it learns to call the find_best_pass_sequence tool to search for a better solution.
> > >
> > >     Learning this complex, conditional logic—knowing when to call which tool and how to act on its output—is a sophisticated policy learning task that requires the capacity of a large model. The "simple" text output is the surface-level manifestation of this learned, robust decision-making policy.
> > >
> > > - **The Clear Impact of Scale:** As you noted from our main figure, model scale is crucial. Our results in Table 1 empirically confirm this: performance steadily increases from GRPO-1.5B (3.87%) to GRPO-3B (5.12%) and finally to GRPO-7B (8.46%). This demonstrates that larger models possess superior reasoning abilities to navigate the optimization space more effectively, a clear example of emergent capabilities.
> > >
> > > ### **3. Alignment with State-of-the-Art Research**
> > > Our approach aligns with the direction of leading research in this area. For instance, Meta's recent **LLM Compiler [9]** work also highlights the power of foundational models for compiler optimization tasks, citing their ability to holistically understand compiler IR. Our work takes this a step further by empowering the LLM within an agentic framework, transforming it from a passive "understander" into an active "decision-maker" guided by reinforcement learning.
> > >
> > >
> > > Thank you again for your insightful feedback. We hope this detailed explanation addresses your concerns.

---

> > > > ### Comment · Reviewer_EFb8 · 2025-08-01
> > > >
> > > > Thank you, this all seems very reasonable to me. I agree that the coding specific knowledge present in the LLM in this task should be useful for making it predict the correct flags. The only thing that seems surprising is that the chain of thoughts do not seem to use information from previous tool calls except in predicting the next one. I am surprised why RL would not emerge a behavior where the model explicitly tries to reason about what to try next? Might you have any intuition for this?

---

> > > > > ### Author Response · Authors · 2025-08-01
> > > > >
> > > > > Thank you for the thoughtful comment. Indeed, it may appear surprising at first that the model does not explicitly reason over what tool to call next. However, under our end-to-end reinforcement learning setup, the model is trained to receive a final reward based on the outcome of the entire trajectory. Each intermediate action—i.e., every thought and tool invocation—is implicitly shaped by this final reward. Only when each individual reasoning step leads to useful tool usage and ultimately a correct answer will the trajectory be rewarded.
> > > > >
> > > > > Thus, the model learns to prefer thought chains and tool sequences that maximize long-term reward. In practice, this results in the emergence of localized reasoning behaviors—even though the model does not explicitly reason about “what to try next,” it learns to generate better intermediate steps that are statistically associated with successful outcomes. Moreover, because our policy is trained with character-level gradient backpropagation, the model gradually internalizes which types of thoughts and tool usages are most beneficial at each stage, without requiring explicit symbolic reasoning.
> > > > >
> > > > > We believe this implicit optimization process explains why the model behaves rationally over time, even without an explicit module for global planning.

---

> > > > > > ### Comment · Reviewer_EFb8 · 2025-08-04
> > > > > >
> > > > > > Thanks for the response. I think this is a fair argument, though it also probably has to do with how the SFT data is prepared. It may ultimately be worth preparing SFT data with more elaborate reasoning as to the choices for tool calls to elicit more reasoning from the model. I *suspect* this will be useful, and also should help developers using this tool to understand how to better optimize for instruction count themselves. (Will make the model less of a black box)
> > > > > >
> > > > > > Thanks again to the authors for detailed responses to my questions, I will raise my score accordingly. Nice work!

---

> > > > > > > ### Author Response · Authors · 2025-08-04
> > > > > > >
> > > > > > > We greatly appreciate your insightful comments and thoughtful suggestions. We wholeheartedly agree with the point raised regarding the importance of SFT data preparation.
> > > > > > >
> > > > > > > We are committed to exploring the construction of more sophisticated and reasoned SFT datasets in our future work, aiming to enhance model interpretability and performance. Thank you once again for your careful review and constructive feedback.

---

### Official Review · Reviewer_9h7w · 2025-06-30

**Clarity:** 3
**Significance:** 3
**Originality:** 3
**Rating:** 4
**Confidence:** 4

**Summary:**

This paper introduces Compiler-R1, a tool assisted system that uses RL to learn better compiler optimization sequences. Instead of relying on fixed optimization strategies, the authors frame the problem as a decision-making task, where an agent selects a sequence of optimization passes tailored to a specific program. The contributions include construction of a high-quality reasoning dataset and a novel two-stage RL pipeline which is first trained with SFT and then improved using tool assisted feedback based on how well the resulting code is optimized in terms of reducing the number of intermediate instructions. The authors evaluate Compiler-R1 on a diverse benchmark of programs and show that it consistently outperforms traditional methods and existing tools in reducing the number of IR. They also study how different ways of representing the input program affect the agent’s performance and perform ablations on how it is important to do both SFT and RL.

**Questions:**

1. The environment relies on external tools like instrcount and find_best_pass_sequence, which may introduce non-determinism or fragility (e.g., if tool crashes or gives misleading feedback) Please describe how errors or noise in tool execution are handled. Are they dropped, or is the agent penalized?
2. SFT models are evaluated with N=40 samples, whereas RL agents use single interaction traces. Could you include a comparison with matched performance (e.g., how many SFT samples are needed to match GRPO-1.5B’s performance),or evaluate both SFT and RL under the same inference budge.
3. The study focuses on LLVM 10 and specific passes. While sufficient for benchmarking, this may limit real-world relevance. Can the system adapt to newer LLVM versions (e.g., v17)?
4. Currently, all programs are treated equally during training. This could slow down learning or introduce instability early in RL training.
Have the authors considered curriculum learning, e.g., starting with short or low-complexity programs? Even a brief analysis of performance across complexity levels could reveal useful insights here.
5. The paper focuses solely on IR instruction count as a proxy for optimization. it's unclear how well this correlates with runtime performance or power/energy efficiency.

**Ethical Concerns:**

["NO or VERY MINOR ethics concerns only"]

**Final Justification:**

I am maintaining my score of 4 (Borderline Accept) after carefully considering the authors’ rebuttal. The paper is technically solid and brings an interesting application of LLMs with RL to compiler optimization, a relatively unexplored domain. The rebuttal provided clarifications on error handling in the RL loop and outlines convergence and penalization strategies. The current reward design lacks multi-objective consideration, the system is evaluated on IR count across benchmarks, but there’s no analysis of generalization to runtime or across complexity levels. The paper contributes an interesting and timely application but falls short on broader generalization and deep technical novelty.

**Limitations:**

The authors have briefly discussed the potential negative impacts of their work including how they can use LLMs to improve interpretability of auto tuning processes and further develop agents to exploit synergies between compiler passes. They have not gone into the details of the societal impact of this work, especially if there is a correctness aspect of discovering the compiler passes, if not properly sandboxed or verified, this could lead to vulnerabilities in deployed software. failures in learned optimization sequences (e.g., those introducing subtle bugs or non-deterministic behavior) could lead to safety risks especially in critical domains.

**Quality:**

3

**Strengths And Weaknesses:**

Strengths:
- As noted by the authors, Compiler-R1 is the first framework to integrate LLMs with RL for automatically tuning compiler pass sequences to minimize LLVM IR instruction count. This makes it an interesting application of tool assisted SFT+RL reasoning in a new domain. The authors also built a reasoning dataset designed for compiler pass optimization.
- The system delivers practical improvements: it reduces LLVM IR instruction count by about 8.5% compared to the standard -Oz optimization. they evaluate on a bunch of different datasets in Compiler Gym and against classical compiler-tuning methods, including Open Tuner, GA, TPE, RIO, Comp Tuner, BOCA, and Auto Phase, showing robustness.

Weaknesses:
- There is not much novelty in terms of training, this paper is mainly applying standard SFT+RL training techniques to a new domain. It would be interesting to see more exploration of tree search, memory-augmented, or curriculum learning for compiler pass discovery.
- Reward design is not deep: The paper optimizes only for IR instruction count reduction. This does not necessarily reflect runtime, energy use, or correctness guarantees. It is not clear how optimizing for IR instruction would generalize to diverse compiler objectives or hardware targets. There's no multi-objective formulation or discussion of real-world utility tradeoffs.
- It would be nice to see more details about the RL training pipeline, like how many steps are needed for convergence; some examples of failure analysis of the tool-assisted pipeline.

---

> ### Author Rebuttal · Authors · 2025-07-30
>
> Thank you very much for your time and effort in reviewing our paper. We sincerely appreciate your thorough feedback and insightful suggestions. Below, we respectfully provide our detailed responses to address your concerns.
>
> ---
>
> ### **W1: Novelty in terms of training**
>
> We appreciate the reviewer’s recognition of our work. While the combination of SFT followed by RL is not novel in isolation, the novelty in our paper lies in the integration of these techniques within the specific domain of **compiler auto-tuning**, a domain where such methods have not been explored previously. Traditional methods for compiler pass optimization rely on **fixed heuristics** or simple rule-based systems. Our framework leverages **RL-driven LLMs**, moving beyond static strategies to **adaptive, autonomous, context-aware optimizations**.
>
> The **novel contributions** of our work can be summarized as:
>
> - **Creation of a high-quality reasoning dataset**: Our dataset is specifically curated for **compiler auto-tuning** tasks, combining **Chain-of-Thought (CoT) reasoning** and **tool-assisted feedback** for effective LLM training. This dataset represents the first attempt to build a comprehensive reasoning dataset for this domain.
> - **First end-to-end RL framework for compiler pass optimization**: While the combination of SFT and RL has been applied in other areas, our work is the first to integrate **RL with LLMs** in **compiler auto-tuning**tasks, offering an innovative framework to **autonomously discover optimized pass sequences** beyond static or heuristic-based methods.
>
> We believe that the **combination of these components** provides a fresh perspective and contributes significantly to **the advancement of compiler optimization**. To address your suggestion, we will make the **novelty of our work** more explicit in the revised manuscript.
>
> ---
>
> ### **W2: Reward design is not deep**
>
> Thank you for highlighting this aspect. While **IR instruction count** serves as a useful proxy for optimization and is widely used in compiler optimization research, we fully acknowledge the importance of a **broader reward design** for real-world relevance.
>
> - **Current reward design**: Our choice of **IR instruction count** is intentional due to its **deterministic** nature and **wide acceptance** in compiler research. It provides a clear, reproducible metric for assessing the optimization quality of the agent’s output.
> - **Future plans for multi-objective optimization**: We recognize the importance of **multi-objective optimization**, and as part of ongoing work, we are exploring how to integrate additional metrics such as **runtime**, **energy consumption**, and **code correctness** into our framework. These can be easily incorporated by **adjusting the reward function** and replacing the current tools with ones that track the desired metrics.
>
> By incorporating these objectives, we aim to improve the real-world **applicability** of **Compiler-R1** and provide a more **comprehensive optimization framework**.
>
> ---
>
> ### **W3: Details about the RL training pipeline**
>
> We appreciate your request for further details about the training pipeline. We are happy to provide more details of the **two-stage training** process. As outlined in **Figure 2,** we first use **SFT** to initialize the model’s reasoning ability. This is followed by **RL**, where the model further refines its strategy based on feedback from interactions with the compilation environment.
>
> Key aspects of the RL training include:
>
> - **RL algorithms**: We use **GRPO**, **PPO**, and **RPP** to fine-tune the model. These methods are applied to iteratively improve the model’s policy, enabling more efficient optimization strategies.
> - **Training Convergence**: Our RL training generally converges within **40 steps**, showcasing the **efficiency** of the pipeline. We will provide **convergence metrics** and additional details on the **training dynamics** in the revised manuscript.
>
> We will also include a more **detailed analysis of the training dynamics**, and other important metrics in our revised manuscript.
>
> ---
>
> ### **Q1: How errors or noise in tool execution are handled**?
>
> Thank you for raising this crucial point. The handling of errors and noise in tool execution is a vital component of ensuring the robustness of **Compiler-R1**.
>
> Our approach to error handling is as follows:
>
> - **Error Detection**: If a tool (e.g., `instrcount`) encounters an invalid pass sequence, it returns an **"error" status**. The model is trained to recognize this and take appropriate corrective action.
> - **Penalization for errors**: Instead of ignoring errors, the **RL reward function** penalizes any trajectory that leads to errors or tool failures. This **negative feedback** helps the model avoid repeating mistakes and promotes more **robust interactions** in future iterations.
>
> Rather than dropping errors, we treat them as **negative feedback** and use them to **improve the robustness** of the agent’s decision-making process. We will add a more detailed explanation of this error-handling mechanism in the final version paper.
>
> ------
>
> ### **Q2: Comparison of matched performance between SFT and RL**
>
> We acknowledge the importance of a fair comparison between **SFT** and **RL** models. To address the concern, we refer to **Figure 3** in our paper, which shows that RL models significantly outperform SFT models:
>
> - **RL models** (e.g., **GRPO-1.5B**) achieve higher **OverOz%** improvement compared to **SFT models** when both are allowed only a **single interaction trace** (as shown in **Figure 3**, at **N attempt** = 1), demonstrating the effectiveness of RL-based exploration.
> - As highlighted in **Figure 3**, the **SFT models have already converged**, yet they still underperform compared to RL models with just one interaction round, indicating that **further gains from SFT are unlikely**, while RL continues to improve through active exploration.
>
> In summary, our results highlight the **clear advantages** of **RL models** over **SFT models**, achieving better optimization with fewer interactions. As shown in **Figure 3**, **RL-based exploration** is more efficient and effective in optimizing compiler pass sequences, underscoring the value of **RL** in complex tasks like compiler tuning.
>
> ------
>
> ### **Q3: Can the system adapt to newer LLVM versions (e.g., v17)?**
>
> Thank you for your query regarding **forward compatibility** with newer LLVM versions. We are confident that Compiler-R1 can be adapted to **newer LLVM versions**, such as LLVM v17, with minimal effort.
>
> Here’s how we ensure **adaptability**:
>
> - **Version-Agnostic Principles**: The fundamental "phase-ordering problem" we solve persists and even grows in complexity with newer compilers. Our framework's core value—learning program-specific strategies to outperform standard heuristics like -Oz—remains highly relevant and is not tied to a specific LLVM version.
> - **Modular design**: Our framework is designed for modularity. To adapt to a newer version like LLVM 17, the only required step is to **update the underlying toolchain**. We would simply replace the opt binary in our execution environment with the new one. The core learning pipeline remains unchanged and can immediately leverage the updated compiler, including its new passes and improvements, without modification.
>
> We will **emphasize the modularity** and **discuss the adaptability** of our framework in the final version paper to highlight these strengths.
>
> ---
>
> ### **Q4: Consideration of curriculum learning**
>
> We appreciate the suggestion to explore **curriculum learning**. While our use of **AutoPhase features** mitigates some of the complexity variations across programs, we acknowledge that **curriculum learning** could be useful, particularly in early RL training stages.
>
> - **Current approach**: The **AutoPhase features** provide a **uniform input representation**, reducing the need for curriculum learning.
> - **Future work**: We plan to investigate the use of **curriculum learning** to better handle **low-complexity programs** in future experiments.
>
> We will include this idea as a **future direction** in the revised manuscript.
>
> ---
>
> ### **Q5: IR instruction count as a proxy for optimization**
>
> Thank you for raising this important point. We agree that IR instruction count as a sole metric has limitations. We chose it because it is a **deterministic, fast-to-measure, and standard proxy** in compiler research, enabling rapid and reproducible experiments.
>
> However, a key strength of our framework is its **flexibility regarding the optimization objective**. Compiler-R1 is not fundamentally tied to IR count and can be readily adapted to target metrics like runtime or power efficiency. This adaptation requires three straightforward changes:
>
> - **Re-construct the Dataset:** Generate new target sequences for the SFT data based on the new objective (e.g., best runtime).
> - **Update the Reward Signal & Tools:** Replace the instrcount tool with a runtime measurement tool and adjust the RL reward accordingly.
> - **Modify the LLM's Prompt:** Change the agent's instructions to reflect the new goal (e.g., "minimize execution time").
> The core agentic framework and its two-stage training pipeline remain unchanged, highlighting its versatility.
>
> We acknowledge the importance of real-world performance, and in **future work**, we plan to extend our framework to include **multi-objective optimization**, incorporating **runtime and energy efficiency** alongside instruction count.
>
> ---
>
> We hope these clarifications may help to address your concerns. At last, we sincerely appreciate your valuable feedback, and we will carefully consider all your suggestions to further improve our paper. We would be deeply grateful if you could kindly reconsider raising the score. Thank you very much!
>
> ---

---

> > ### Comment · Reviewer_9h7w · 2025-08-03
> >
> > Thanks to the authors for providing a detailed response to each of my points, I would like to leave the rating unchanged.

---

> > > ### Author Response · Authors · 2025-08-05
> > >
> > > Thank you for your kind words! We greatly appreciate your engagement with our work and the helpful suggestions you provided.
> > >
> > > We will take great care in ensuring the final version meets your expectations. Once again, thank you for your thorough review and constructive feedback!

---

### Official Review · Reviewer_Wbgx · 2025-07-01

**Clarity:** 3
**Significance:** 3
**Originality:** 3
**Rating:** 5
**Confidence:** 3

**Summary:**

The paper proposes a LLM-based algorithmic framework (Compiler-R1) for program optimization. The primary contributions of the paper are algorithmic and empirical. Specifically, the system uses a two-stage training pipeline (SFT followed by RL) to train a LLM to generate an optimized sequence of compiler operations (passes) by leveraging standard compiler optimization and program analysis tools in order to reduce the instruction count of the compiled program. The paper proposes a method to construct a dataset for the Stage 1 (SFT) pipeline. Experiments on CompilerGym tasks show that Compiler-R1 is reasonably effective at reducing instruction size while running faster than strong baselines.

**Questions:**

1. Please enumerate the novel contributions of the paper wrt the dataset, algorithmic ideas and empirical (instruction count) improvements over previous SOTA on CompilerGym.

2. How well does Compiler-R1 generalize to programs outside its training distribution? E.g., large programs.

**Ethical Concerns:**

["NO or VERY MINOR ethics concerns only"]

**Final Justification:**

I thank the authors for their detailed response, which have alleviated my main concerns about the approach. After reading the other reviews and comments, I continue to recommend acceptance of the paper, maintaining my current score.

**Limitations:**

- Somewhat. The paper could do a better job of discussing ways in which the empirical results might be invalid or performance might drop. Examples include cases instances where Compiler-R1 might fail to generalize due to OOD inputs as well as the potential for test dataset leakage during training of the base LLM.

**Paper Formatting Concerns:**

None.

**Quality:**

3

**Strengths And Weaknesses:**

**Strengths**

- The paper studies an interesting and important problem. Automated improvements to program compilation tasks is an important area and progress here is likely to have significant impact.

- The paper is well written. Key ideas are explained clearly and I found it easy to follow. The paper appendix includes a good amount of empirical details, which, in conjunction with released code and datasets, will aid reproducibility.

- The proposed algorithmic approach is intuitively clear. The two-stage pipeline (SFT followed by RL with GRPO) using AutoPhase features makes a lot of sense. The dataset construction approach for the SFT stage seems like a standalone contribution in itself (as far as I can tell). While the algorithmic novelty is a bit low in that the components are all off-the-shelf, the combination of existing ideas is novel, to my knowledge. (The paper could do a better job of making its novel contributions clearer.)

- The experiments show that Compiler-R1 is generally effective at reducing instruction counts while running faster than most baselines. It strikes a nice balance between AutoPhase (higher instruction count but fast) and Opentuner (lower instruction count but slower). The experiments are richly detailed and contain a good amount of insight into the underlying factors driving performance gains. In particular, the importance of RL training and scaling model size comes through clearly.

**Weaknesses**

- The paper could do a better job of describing its novelty and contributions relative to the larger body of work on compiler optimization. For example, how novel is the induced dataset generated for the SFT training to the community? How important or impactful is the improvement over traditional autotuners in Table 1 (8.46 vs 6.09)? A more detailed discussion of the novelty and impact aspects of the dataset and empirical results (perhaps in the appendix) would significantly improve the paper.

- The paper would be strengthened with a detailed discussion of Compiler-R1's generalization to programs outside CompilerGym (i.e., OOD). For example, what would happen on larger programs (> 10K instruction count which were filtered out) or those significantly different from the tasks in CompilerGym? Put differently, in what scenarios might Compiler-R1 fail?

- The above are relatively minor. Overall, I didn't find any major issues in the paper and I'm open to increasing my score with a better understanding of the empirical contributions.

---

> ### Author Rebuttal · Authors · 2025-07-30
>
> Thank you very much for your time and effort in reviewing our paper. We sincerely appreciate your thorough feedback and insightful suggestions. Below, we respectfully provide our detailed responses to address your concerns.
>
> ---
>
> ### W1 & Q1: Contributions of the paper wrt the dataset, algorithmic ideas and empirical improvements over previous SOTA on CompilerGym.
>
> Thank you for this question. We are happy to clarify our novel contributions:
>
> * **On the Dataset**: We present the **first high-quality, reasoning-centric dataset** tailored specifically for **compiler auto-tuning tasks**. Our key innovation lies in the systematic methodology used for constructing the dataset:
>
>   - We curate a **global knowledge graph** by identifying **synergistic pass pairs**, which then serve to guide the generation of optimal pass sequences.
>   - We introduce a **Chain-of-Thought (CoT) framework**, teaching the model not only what the answer is but also how to reason and use tools effectively to derive it.
>
>   This structured approach addresses the long-standing lack of reasoning data in complex, tool-intensive domains like compiler auto-tuning.
>
> * **On Algorithmic Ideas**: The core innovation of our work is the introduction of **Compiler-R1**, **the first LLM-based agent specifically designed for compiler auto-tuning**. While the two-stage approach of SFT followed by RL is not novel in isolation, our **application of this combination within the domain of compiler optimization** is the novelty. Traditional approaches to compiler pass optimization typically rely on **heuristics** or **static strategies**, whereas our agent uses **interactive reinforcement learning** to autonomously refine optimization strategies, thus moving beyond imitation learning. This makes **Compiler-R1 the first interactive agent** in this domain, marking a significant leap forward.
>
> - **On Empirical Improvements**: Our results demonstrate **state-of-the-art performance** for **LLM-based auto-tuning**. Specifically:
>   - **Compiler-R1** (GRPO-7B) achieves an **8.46% average reduction** in IR instruction count compared to the **opt -Oz** baseline. This performance **significantly surpasses** the **strongest SFT-only baseline** (4.66%) and **traditional autotuners** like **OpenTuner** (6.09%).
>   - We also highlight **the computational efficiency** of our framework: Compiler-R1 completes the task in **~26 seconds**, much faster than traditional methods that can take over **200 seconds**.
>
> These contributions not only stablish **Compiler-R1** as a leading framework in **LLM-driven compiler auto-tuning**, but also provide a blueprint for creating **interactive agents** capable of solving complex, tool-assisted tasks in diverse domains.
>
> ------
>
>
> ### W2 & Q2: How well does Compiler-R1 generalize to programs outside its training distribution?
>
> Thank you for this excellent question, which addresses a critical aspect of our work. Our design choices have been made to ensure Compiler-R1 is inherently more generalizable, even to large programs, and we believe our approach offers strong potential for generalization outside of its training distribution.
>
> #### **1. Rationale for Focusing on Programs Under 10k Instructions**
>
> First, it’s important to clarify our decision to focus on programs with fewer than 10k instructions. This was primarily driven by the need for **a fair and feasible comparison** with traditional auto-tuning baselines such as **BOCA** and **CompTuner**. These methods often require extensive time to tune even a single moderate-sized program, and the time cost becomes prohibitive when scaling to larger programs. Limiting the size to **<10k instructions** allowed us to perform a **rigorous and direct comparison** across all methods, ensuring the **comparability** of results. However, we acknowledge the importance of **testing on larger programs** and are exploring this as a next step.
>
> #### **2. Generalization to Larger Programs with AutoPhase Features**
>
> Second, **Compiler-R1** is uniquely positioned to handle larger programs, thanks to its use of a **fixed-size, 56-dimensional AutoPhase feature vector**. This design choice has profound implications for **generalization** and scalability, offering two key advantages:
>
> - **Decoupling Program Size from Input Size**: Whether a program has **1,000 instructions** or **1,000,000 instructions**, it is represented by the same compact **56-element vector**. The program’s size only affects the numerical values within this vector, meaning the model does not rely on processing raw code directly. This decoupling **bypasses context window limitations** that typically hinder models based on raw code representation, enabling **better scalability**.
> - **Encouraging Scale-Invariant Learning**: The model learns to make decisions based on **structural patterns** and **proportional relationships** reflected in the features, such as the **ratio of branch instructions to memory instructions**. These structural signatures tend to be shared across programs of different sizes. By focusing on the **"shape"** of the program rather than its absolute length, the model can **learn generalizable optimization heuristics**, which makes it more robust to varying program sizes and complexities.
>
> In essence, the task of generalization is simplified from understanding **arbitrary-length code** to **mapping points within a structured, low-dimensional feature space**. This approach inherently promotes **generalizable and scalable learning**, and while we acknowledge the need for **empirical validation** on larger program benchmarks, we are confident that this feature representation will provide a **strong foundation** for **out-of-distribution robustness**.
>
> We will add this clarification to our paper to emphasize how our **feature-based design** enables **generalization** and why we believe it positions **Compiler-R1** well for scalability to larger programs.
>
> ---
>
> We hope this response addresses your concerns and demonstrates the novelty and robustness of **Compiler-R1**. We truly value your feedback and will incorporate your suggestions to further improve our paper. Thank you once again for your careful and constructive review!
>
> ---

---

> ### Comment · Reviewer_Wbgx · 2025-08-05
> **Re. author response**
>
> I thank the authors for their detailed response, which have alleviated my main concerns about the approach. After reading the other reviews and comments, I continue to recommend acceptance of the paper, maintaining my current score.

---

> > ### Author Response · Authors · 2025-08-05
> >
> > Thank you for your kind words! We greatly appreciate your engagement with our work and the helpful suggestions you provided.
> >
> > We will take great care in ensuring the final version meets your expectations. Once again, thank you for your thorough review and constructive feedback!

---

### Note · Authors · 2025-08-13

We sincerely thank the reviewers for their insightful feedback. Below, we summarize our responses to the major concerns raised during the review process:

1. **Generalization to Larger Programs (>10k instructions)**: Our focus on smaller programs was to ensure comparability with existing baselines. However, **AutoPhase features enable scalability** by decoupling program size from the input representation. This compact feature vector is invariant to program length, supporting generalization and **ensuring that Compiler-R1 can handle larger programs effectively**.
2. **Empirical Comparisons and Performance**: While Compiler-R1 models leverage LLMs, which are computationally intensive, they benefit from parallelization and fast GPU inference. Traditional autotuners, though computationally less expensive per pass, require sequential evaluations that result in longer runtimes. **Compiler-R1 offers a compelling trade-off between optimization quality and computational efficiency**.
3. **Reward Design and Multi-Objective Optimization**: We currently focus on IR instruction count reduction due to its **simplicity and determinism**, widely accepted in compiler research. However, we plan to extend our framework to include **runtime, energy consumption, and correctness,** providing a more holistic evaluation and reflecting real-world utility.
4. **Error Handling and Robustness**: We handle errors from external tools by **penalizing trajectories that lead to tool failures**. This prevents the model from repeating mistakes and encourages more robust behavior, allowing it to adapt and improve its interaction with the compilation environment.
5. **Future Directions**: While AutoPhase mitigates complexity variations, we see value in exploring curriculum learning for lower-complexity programs. We also aim to adapt the framework to newer LLVM versions and integrate additional optimization objectives.

In summary, the key innovations of our work are the development of **the first high-quality reasoning dataset for compiler auto-tuning** and the introduction of Compiler-R1, **an RL-driven framework that allows agents to autonomously discover optimization strategies**, advancing beyond heuristic approaches and highlighting the potential of RL-based agents in system optimization tasks.

Once again, we sincerely appreciate the detailed and constructive feedback provided, which has been invaluable in refining our work. Thank you for your time and consideration.

---

### Decision · Program_Chairs · 2025-09-17

**Decision:**

Accept (poster)

**Comment:**

The paper designs and studies a compiler optimization approach based on using LLMs with the SFT+RL paradigm.

This is a strong application-oriented contribution and could influence the compiler optimization community. It is also a non-trivial instantiation of the SFT+RL paradigm that could be beneficial as a schema for researchers with analogous problems.

There was significant interaction with the authors during the rebuttal phase and most concerns were address. The major concern included:

- Limited insights into why we are seeing improvements given that there are few options available to the model.

- Questioning the necessity of the LLM here compared to something simpler.

While these questions are not fully resolved for all reviewers, the authors have indicated solid ways they can adjust the paper to improve in those directions.

Overall, this is a solid contribution.